# How do Language Models Bind Entities in Context?

**Jiahai Feng**[*] **& Jacob Steinhardt**
UC Berkeley

## Abstract

To correctly use in-context information, language models (LMs) must bind entities to their attributes. For example, given a context describing a "green square" and a "blue circle", LMs must bind the shapes to their respective colors. We analyze LM representations and identify the *binding ID mechanism*: a general mechanism for solving the binding problem, which we observe in every sufficiently large model from the Pythia and LLaMA families. Using causal interventions, we show that LMs' internal activations represent binding information by attaching *binding ID vectors* to corresponding entities and attributes. We further show that binding ID vectors form a continuous subspace, in which distances between binding ID vectors reflect their discernability. Overall, our results uncover interpretable strategies in LMs for representing symbolic knowledge in-context, providing a step towards understanding general in-context reasoning in large-scale LMs.

## 1 Introduction

Modern language models (LMs) excel at many reasoning benchmarks, suggesting that they can perform general purpose reasoning across many domains. However, the mechanisms that underlie LM reasoning remain largely unknown (Räuker et al., 2023). The deployment of LMs in society has led to calls to better understand these mechanisms (Hendrycks et al., 2021), so as to know why they work and when they fail (Mu & Andreas, 2020; Hernandez et al., 2021; Vig et al., 2020b).

In this work, we seek to understand *binding*, a foundational skill that underlies many compositional reasoning capabilities (Fodor & Pylyshyn, 1988) such as entity tracking (Kim & Schuster, 2023). How humans solve binding, i.e. recognize features of an object as bound to that object and not to others, is a fundamental problem in psychology (Treisman, 1996). Here, we study binding in LMs.

Binding arises any time the LM has to reason about two or more objects of the same kind. For example, consider the following passage involving two people and two countries:

> Context: Alice lives in the capital city of France. Bob lives in the capital city of Thailand.
> Question: Which city does Bob live in?       (1)

In this example the LM has to represent the associations *lives(Alice, Paris)* and *lives(Bob, Bangkok)*. We call this the *binding problem*—for the predicate *lives*, *Alice* is bound to *Paris* and *Bob* to *Bangkok*. Since predicates are bound in-context, binding must occur in the activations, rather than in the weights as with factual recall (Meng et al., 2022). This raises the question: how do LMs represent binding information in the context such that they can be later recalled?

Overall, our key technical contribution is the identification of a robust general mechanism in LMs for solving the binding problem. The mechanism relies on *binding IDs*, which are abstract concepts that LMs use internally to mark variables in the same predicate apart from variables in other predicates (Fig. 1). Using causal mediation analysis we empirically verify two key properties of the binding ID mechanism (§3): factorizability and position independence.

Turning to the structure of binding IDs, we find that binding IDs are represented as vectors which are bound to variables by simple addition (§4) in the activation space. Further, we show that binding IDs occupy a subspace, in the sense that linear combinations of binding IDs are still valid binding IDs, even though random vectors are not.

---

[*]Correspondence to fjiahai@berkeley.edu

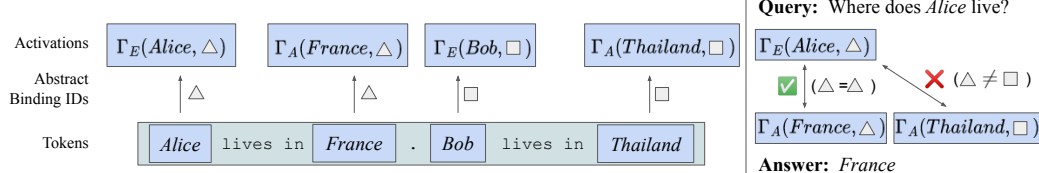

Figure 1: The Binding ID mechanism. The LM learns abstract binding IDs (drawn as triangles or squares) which distinguish between entity-attribute pairs. Binding functions $\Gamma_E$ and $\Gamma_A$ bind entities and attributes to their abstract binding ID, and store the results in the activations. To answer queries, the LM identifies the attribute that shares the same binding ID as the queried entity.

Lastly, we find that binding IDs are ubiquitous and transferable (§5). They are used by every sufficiently large model in the LLaMA (Touvron et al., 2023) and Pythia (Biderman et al., 2023) families, and their fidelity increases with scale. They are used for a variety of synthetic binding tasks with different surface forms, and binding vectors from one task transfer to other tasks. Finally, we qualify our findings by showing that despite their ubiquity, binding IDs are not universal: we exhibit a question-answering task where an alternate mechanism, "direct binding", is used instead (§E). We release code and datasets here: https://github.com/jiahai-feng/binding-iclr

## 2 PRELIMINARIES

In this section we define the *binding task* and explain causal mediation analysis, our main experimental technique.

**Binding task.** Formally, the binding task consists of a set of entities $\mathcal{E}$ and a set of attributes $\mathcal{A}$. An $n$-entity instance of the binding task consists of a context that is constructed from $n$ entities $e_0, \ldots, e_{n-1} \in \mathcal{E}$ and $n$ attributes $a_0, \ldots, a_{n-1} \in \mathcal{A}$, and we denote the corresponding context as $\mathbf{c} = \text{ctxt}(e_0 \leftrightarrow a_0, \ldots, e_{n-1} \leftrightarrow a_{n-1})$. For a context $\mathbf{c}$, we use $E_k(\mathbf{c})$ and $A_k(\mathbf{c})$ to denote the $k$-th entity and the $k$-th attribute of the context $\mathbf{c}$, for $k \in [0, n-1]$. We will drop the dependence on $\mathbf{c}$ for brevity when the choice of $\mathbf{c}$ is clear from context.

In the CAPITALS task, which is the main task we study for most of the paper, $\mathcal{E}$ is a set of single-token names, and $\mathcal{A}$ is a set of single-token countries. Quote 1 is an example instance of the CAPITALS task with context $\mathbf{c} = \text{ctxt}(Alice \leftrightarrow France, Bob \leftrightarrow Thailand)$. In this context, $E_0$ is $Alice$, $A_0$ is $France$, etc.

Given a context $\mathbf{c}$, we are interested in the model's behavior when queried with each of the $n$ entities present in $\mathbf{c}$. For any $k \in [0, n-1]$, when queried with the entity $E_k$ the model should place high probability on the answer matching $A_k$. In our running example, the model should predict "Paris" when queried with "Alice", and "Bangkok" when queried with "Bob".

To evaluate a model's behavior on a binding task, we sample $N = 100$ contexts. For each context $\mathbf{c}$, we query the LM with every entity mentioned in the context, which returns a vector of log probabilities over every token in the vocabulary. The *mean log prob* metric measures the mean of the log probability assigned to the correct attribute token. Top-1 accuracy measures the proportion of queries where the correct attribute token has the highest log probability out of all attribute tokens. However, we will instead use the *median-calibrated accuracy* (Zhao et al., 2021), which calibrates the log probabilities with the median log probability before taking the top-1 accuracy. We discuss this choice in §A.

**Causality in autoregressive LMs.** We utilize inherent causal structure in autoregressive LMs. Let an LM have $n_{\text{layers}}$ transformer layers and a $d_{\text{model}}$-dimensional activation space. For every token position $p$, we use $Z_p \in \mathbb{R}^{n_{\text{layers}} \times d_{\text{model}}}$ to denote the stacked set of internal activations[1] at token $p$ (see Fig. 2a). We refer to the collective internal activations of the context as $Z_{\text{context}}$. In addition, we denote the activations at the token for the $k$-th entity as $Z_{E_k}$, and the $k$-th attribute as $Z_{A_k}$. We sometimes write $Z_{A_k}(\mathbf{c}), Z_{\text{context}}(\mathbf{c})$, etc. to make clear the dependence on the context $\mathbf{c}$.

---

[1]These are the pre-transformer layer activations, sometimes referred to as the *residual stream*.

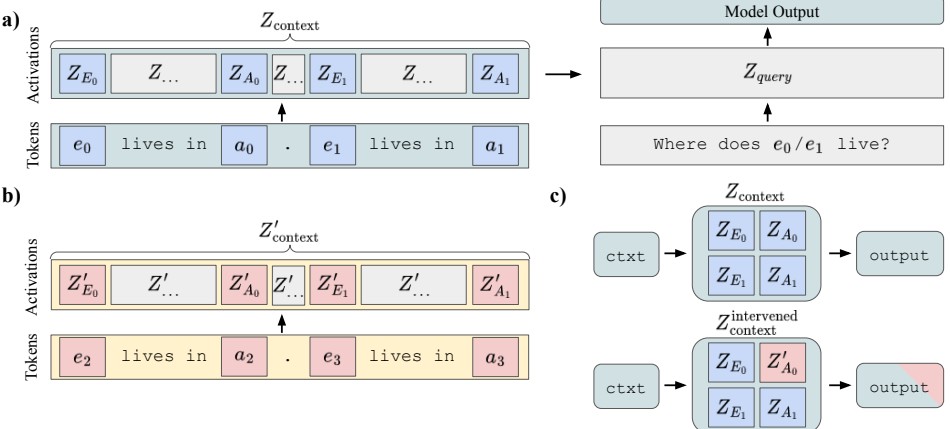

Figure 2: **a)** Causal diagram for autoregressive LMs. From input context $\mathrm{ctxt}(e_0 \leftrightarrow a_0, e_1 \leftrightarrow a_1)$, the LM constructs internal representations $Z_{\mathrm{context}}$. We will mainly study the components of $Z_{\mathrm{context}}$ boxed in blue. **b)** A secondary run of the LM on context $\mathrm{ctxt}(e_2 \leftrightarrow a_2, e_3 \leftrightarrow a_3)$ to produce $Z'_{\mathrm{context}}$. **c)** An example intervention where $Z_{\mathrm{context}}$ is modified by replacing $Z_{A_0} \to Z'_{A_0}$ from $Z'_{\mathrm{context}}$.

Fig. 2a shows that $Z_{\mathrm{context}}$ contains all the information about the context that the LM uses. We thus study the structure of $Z_{\mathrm{context}}$ using *causal mediation analysis*, a widely used tool for understanding neural networks (Vig et al., 2020a; Geiger et al., 2021; Meng et al., 2022). Causal mediation analysis involves substituting one set of activations in a network for another, and we adopt the $/.$ notation (from Mathematica) to denote this. For example, for activations $Z_* \in \mathbb{R}^{n_{\mathrm{layers}} \times d_{\mathrm{model}}}$, and a token position $p$ in the context, $Z_{\mathrm{context}}/.\{Z_p \to Z_*\} = [Z_0, \ldots, Z_{p-1}, Z_*, Z_{p+1}, \ldots]$. Similarly, for a context $\mathbf{c} = \mathrm{ctxt}(e_0 \leftrightarrow a_0, \ldots, e_{n-1} \leftrightarrow a_{n-1})$, we have $\mathbf{c}/.\{E_k \to e_*\} = \mathrm{ctxt}(e_0 \leftrightarrow a_0, \ldots, e_* \leftrightarrow a_k, \ldots, e_{n-1} \leftrightarrow a_{n-1})$.

Given a causal graph, causal mediation analysis determines the role of an intermediate node by experimentally intervening on the value of the node and measuring the model's output on various queries. For convenience, when the model answers queries in accordance to a context $\mathbf{c}$, we say that the model *believes*[2] $\mathbf{c}$. If there is no context consistent with the language model's behavior, then we say that the LM is *confused*.

As an example, suppose we are interested in the role of the activations $Z_{A_0}$ in Fig. 2a. To apply causal mediation analysis, we would:

1. Obtain $Z_{\mathrm{context}}$ by running the model on the original context $\mathbf{c}$ (which we also refer to as the *target* context) (Fig. 2a)

2. Obtain $Z'_{\mathrm{context}}$ by running the model on a different context $\mathbf{c}'$ (i.e. *source* context) (Fig. 2b)

3. Modify $Z_{\mathrm{context}}$ by replacing $Z_{A_0}$ from the target context with $Z'_{A_0}$ from the source context (Fig. 2c), while keeping all other aspects of $Z_{\mathrm{context}}$ the same, resulting in $Z^{\mathrm{intervened}}_{\mathrm{context}} = Z_{\mathrm{context}}/.\{Z_{A_0} \to Z'_{A_0}\}$

4. Evaluate the model's beliefs based on the new $Z^{\mathrm{intervened}}_{\mathrm{context}}$

We can infer the causal role of $Z_{A_0}$ from how the intervention $Z_{\mathrm{context}}/.\{Z_{A_0} \to Z'_{A_0}\}$ changes the model's beliefs. Intuitively, if the model retains its original beliefs $\mathbf{c}$, then $Z_{A_0}$ has no causal role in the model's behavior on the binding task. On the other hand, if the model now believes the source context $\mathbf{c}'$, then $Z_{A_0}$ contains all the information in the context. In reality both hypothetical extremes are implausible, and in §3 we discuss a more realistic hypothesis.

A subtle point is that we study how different components of $Z_{\mathrm{context}}$ store information about the context (and thus influence behavior), and not how $Z_{\mathrm{context}}$ itself is constructed. We thus suppress

---

[2]We do not claim or assume that LMs actually have beliefs in the sense that humans do. This is a purely notational choice to reduce verbosity.

the causal influence that $Z_{A_0}$ has on downstream parts of $Z_\text{context}$ (such as $Z_{E_1}$ and $Z_{A_1}$) by freezing the values of $Z_{E_1}$ and $Z_{A_1}$ in $Z_\text{context}^\text{intervened}$ instead of recomputing them based on $Z'_{A_0}$.

## 3 EXISTENCE OF BINDING IDS

In this section, we first describe our hypothesized binding ID mechanism. Then, we identify two key predictions of the mechanism, factorizability and position independence, and verify them experimentally. We provide an informal argument in §B for why this binding ID mechanism is the only mechanism consistent with factorizability and position independence.

**Binding ID mechanism.** We claim that to bind attributes to entities, the LM learns abstract binding IDs that it assigns to entities and attributes, so that entities and attributes bound together have the same binding ID (Fig. 1). In more detail, our informal description of the binding ID mechanism is:

1. For the $k$-th entity-attribute pair construct an abstract binding ID that is independent of the entity/attribute values. Thus, for a fixed $n$-entity binding task (e.g. CAPITALS task) we can identify the $k$-th abstract binding ID with the integer $k \in \{0, \dots, n-1\}$.
2. For entity $E_k$, encode both the entity $E_k$ and the binding ID $k$ in the activations $Z_{E_k}$.
3. For attribute $A_k$, encode both the attribute $A_k$ and the binding ID $k$ in the activations $Z_{A_k}$.
4. To answer a query for entity $E_k$, retrieve from $Z_\text{context}$ the attribute that shares the same binding ID as $E_k$.

Further, for activations $Z_{E_k}$ and $Z_{A_k}$, the binding ID and the entity/attribute are the only information they contain that affects the query behavior.

More formally, there are *binding functions* $\Gamma_E(e, k)$ and $\Gamma_A(a, k)$ that fully specify how $Z_E$ and $Z_A$ bind entities/attributes with binding IDs. Specifically, if $E_k = e \in \mathcal{E}$, then we can replace $Z_{E_k}$ with $\Gamma_E(e, k)$ without changing the query behavior, and likewise for $Z_A$.

More generally, given $Z_\text{context}$ with entity representations $\Gamma_E(e_0, 0), \dots, \Gamma_E(e_{n-1}, n-1)$ and attribute representations $\Gamma_A(a_0, \pi(0)), \dots, \Gamma_A(a_{n-1}, \pi(n-1))$ for a permutation $\pi$, the LM should answer queries according to the context $\mathbf{c} = \text{ctxt}(e_0 \leftrightarrow a_{\pi^{-1}(0)}, \dots, e_{n-1} \leftrightarrow a_{\pi^{-1}(n-1)})$. This implies two properties in particular, which we will test in the following subsections:

- **Factorizability:** if we replace $Z_{A_k}$ with $Z_{A'_k}$, then the model will bind $E_k$ to $A'_k$ instead of $A_k$, i.e. it will believe $\mathbf{c}./\{A_k \to A'_k\}$. This is because $Z'_{A_k}$ encodes $\Gamma_A(A'_k, k)$ and $Z_{A_k}$ encodes $\Gamma_A(A_k, k)$. Substituting $Z_{A_k} \to Z_{A'_k}$ will overwrite $\Gamma_A(A_k, k)$ with $\Gamma_A(A'_k, k)$, causing the model to bind $E_k$ to $A'_k$.
- **Position independence:** if we e.g. swap $Z_{A_0}$ and $Z_{A_1}$, the model still binds $A_0 \leftrightarrow E_0$ and $A_1 \leftrightarrow E_1$, because it looks up attributes based on binding ID and not position in the context.

In §4, we construct fine-grained modifications to the activation $Z$ that modify the binding ID but not the attributes, allowing us to test the binding hypothesis more directly. In §5 we extend this further by showing that binding IDs can be transplanted from entirely different tasks.

### 3.1 FACTORIZABILITY OF ACTIVATIONS

The first property of $Z_\text{context}$ we test is *factorizability*. In our claimed mechanism, information is highly localized—$Z_{A_k}$ contains all relevant information about $A_k$, and likewise for $Z_{E_k}$. Therefore, we expect LMs that implement this mechanism to have factorizable activations: for any contexts $\mathbf{c}, \mathbf{c}'$, substituting $Z_{E_k} \to Z_{E_k}(\mathbf{c}')$ into $Z_\text{context}(\mathbf{c})$ will cause the model to believe $\mathbf{c}./\{E_k \to E'_k\}$, and substituting $Z_{A_k} \to Z_{A_k}(\mathbf{c}')$ cause the model to believe $\mathbf{c}./\{A_k \to A'_k\}$.

To test this concretely, we considered the CAPITALS task from §2 with $n = 2$ entity-attribute pairs. We computed representations for two contexts $\mathbf{c} = \text{ctxt}(e_0 \leftrightarrow a_0, e_1 \leftrightarrow a_1)$ and $\mathbf{c}' = \text{ctxt}(e'_0 \leftrightarrow a'_0, e'_1 \leftrightarrow a'_1)$, and used causal mediation analysis (§2) to swap representations from the source context $\mathbf{c}'$ into the target context $\mathbf{c}$. Specifically, we fix $k \in \{0, 1\}$ and intervene on either just the entity ($Z_{E_K} \to Z'_{E_k}$), just the attribute, neither, or both. We then measure the mean log probs for all possible queries ($E_0, E_1, E'_0, E'_1$). For instance, swapping $A_k$ with $A'_k$ in $Z_\text{context}$ should lead $A'_k$ (and not $A_k$) to have high log-probability when $E_k$ is queried.

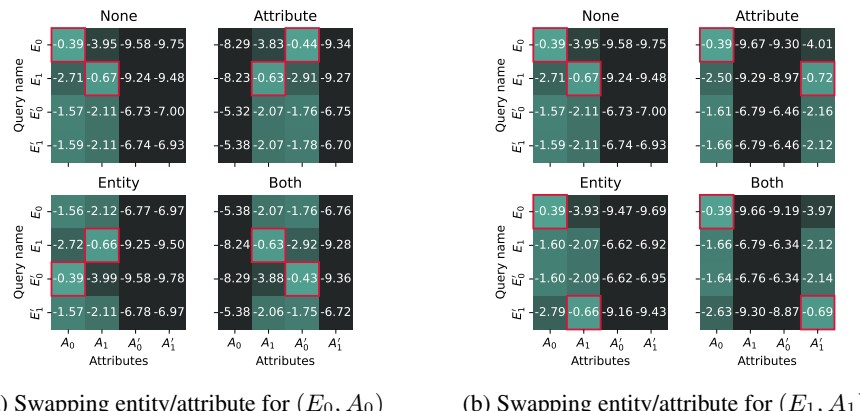

(a) Swapping entity/attribute for $(E_0, A_0)$      (b) Swapping entity/attribute for $(E_1, A_1)$

Figure 3: Factorizability results. Each row corresponds to querying for a particular entity. Plotted are the mean log prob for all four attributes. Highlighted squares are predicted by factorizability.

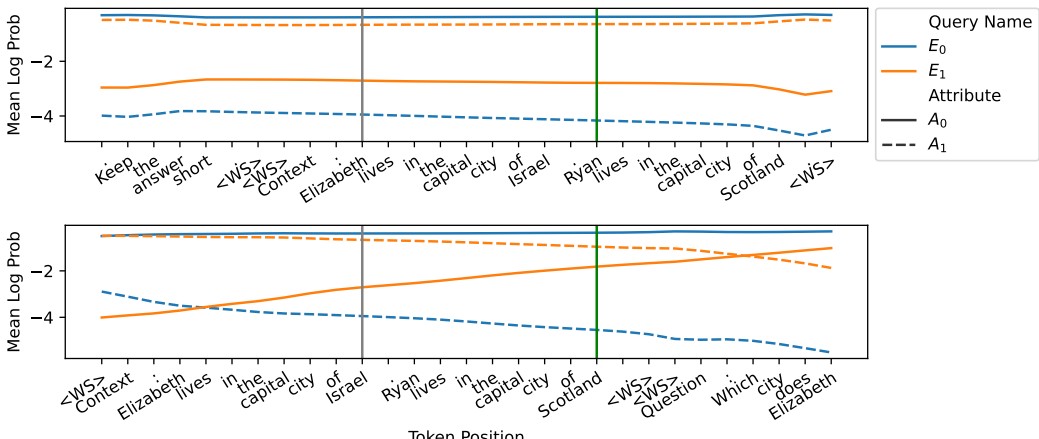

Figure 4: Top: Mean log probs for entity interventions. Bottom: Mean log probs for attributes. For brevity, let $Z_k$ refer to $Z_{E_k}$ or $Z_{A_k}$. The grey and green vertical lines indicate the original positions for $Z_0$ and $Z_1$ respectively. The x-axis marks $x$, $Z_0$'s new position. Under the position interventions $\{X_0 \rightarrow x, X_1 \rightarrow X_1 - (x - X_0)\}$, the grey line is the *control condition* with no interventions, and the green line is the *swapped condition* where $Z_0$ and $Z_1$ have swapped positions.

Results are shown in Fig. 3 and support the factorizability hypothesis. As an example, consider Fig. 3a. In the None setting (no intervention), we see high log probs for $A_0$ when queried for $E_0$, and for $A_1$ when queried for $E_1$. This indicates that the LM is able to solve this task. Next, consider the Attribute intervention setting ($A_0 \rightarrow A_0'$): querying for $E_0$ now gives high log probs for $A_0'$, and querying for $E_1$ gives $A_1$ as usual. Finally, in the Both setting (where both entity and attribute are swapped), querying $E_0'$ returns $A_0'$ while querying $E_0$ leads to approximately uniform predictions.

**Experiment details.** We use LLaMA 30-b here and elsewhere unless otherwise stated. In practice, we found that activations for both the entity token and the subsequent token encode the entity binding information. Thus for all experiments in this paper, we expand the definition of $Z_{E_k}$ to include the token activations immediately after $E_k$.

## 3.2 POSITION INDEPENDENCE

We next turn to position independence, which is the other property we expect LMs implementing the binding ID mechanism to have. This says that permuting the order of the $Z_{E_k}$ and $Z_{A_k}$ should

have no effect on the output, because the LM looks only at the binding IDs and not the positions of entities or attributes activations.

To apply causal interventions to the positions, we use the fact that transformers use positional embeddings to encode the (relative) position of each token in the input. We can thus intervene on these embeddings to "move" one of the $Z_k$'s to another location $k'$. Formally, we let $X_k$ describe the position embedding for $Z_k$, and denote the position intervention as $\{X_k \to k'\}$. In §C we describe how to do this for rotary position embeddings (RoPE), which underlie all the models we study. For now, we will assume this intervention as a primitive and discuss experimental results.

For our experiments, we again consider the CAPITALS task with $n = 2$. Let $X_{E_0}$ and $X_{E_1}$ denote the positions of the two entities. We apply interventions of the form $\{X_{E_0} \to x, X_{E_1} \to X_{E_1} - (x - X_{E_0})\}$, for $x \in \{X_{E_0}, X_{E_0} + 1, \ldots, X_{E_1}\}$. This measures the effect of gradually moving the two entity positions past each other: when $x = X_{E_0}$, no intervention is performed (*control condition*), and when $x = X_{E_1}$ the entity positions are swapped (*swapped condition*). We repeat the same experiment with attribute activations and measure the mean log probs in both cases.

Results are shown in Fig. 4. As predicted under position independence, position interventions result in little change in model behavior. Consider the *swapped condition* at the green line. Had the binding information been entirely encoded in position, we expect a complete switch in beliefs compared to the *control condition*. In reality, we observe almost no change in mean log probs for entities and a small change in mean log probs for attributes that seems to be part of an overall gradual trend.

We interpret this gradual trend as an artifact of *position-dependent bias*, and not as evidence against position independence. We view it as a bias because it affects all attributes regardless of how they are bound—attributes that are shifted to later positions always have higher log probs. We provide further discussion of this bias, as well as other experimental details, in §C.

## 4 STRUCTURE OF BINDING ID

The earlier section shows evidence for the binding ID mechanism. Here, we investigate two hypotheses on the structure of binding IDs and binding functions. The first is that the binding functions $\Gamma_A$ and $\Gamma_E$ are additive, which lets us think of binding IDs as *binding vectors*. The second is contingent on the first, and asks if binding vectors have a geometric relationship between each other.

### 4.1 ADDITIVITY OF BINDING FUNCTIONS

Prior interpretability research has proposed that transformers represent features linearly (Elhage et al., 2021). Therefore a natural hypothesis is that both entity/attribute representations and abstract binding IDs are vectors in activation space, and that the binding function (§3) simply adds the vectors for entity/attribute and binding ID. We let the binding ID $k$ be represented by the pair of vectors $[b_E(k), b_A(k)]$, and the representations of entity $e$ and attribute $a$ be $f_E(e)$ and $f_A(a)$ respectively. Then, we hypothesize that the binding functions can be linearly decomposed as:

$$\Gamma_A(a, k) = f_A(a) + b_A(k), \quad \Gamma_E(e, k) = f_E(e) + b_E(k). \tag{1}$$

Binding ID vectors seem intuitive and plausibly implementable by transformer circuits. To experimentally test this, we seek to extract $b_A(k)$ and $b_E(k)$ in order to perform vector arithmetic on them. We use (1) to extract the *differences* $\Delta_E(k) := b_E(k) - b_E(0)$, $\Delta_A(k) := b_A(k) - b_A(0)$. Rearranging (1), we obtain

$$\Delta_A(k) = \Gamma_A(a, k) - \Gamma_A(a, 0), \quad \Delta_E(k) = \Gamma_E(e, k) - \Gamma_E(e, 0). \tag{2}$$

We estimate $\Delta_A(k)$ by sampling $\mathbb{E}_{\mathbf{c}, \mathbf{c}'}[Z_{A_k}(\mathbf{c}) - Z_{A_0}(\mathbf{c}')]$, and likewise for $\Delta_E(k)$.

**Mean interventions.** With the difference vectors, we can modify binding IDs by performing *mean interventions*, and observe how model behavior changes. The attribute mean intervention switches the binding ID vectors in $Z_{A_0}$ and $Z_{A_1}$ with the interventions $Z_{A_0} \to Z_{A_0} + \Delta_A(1), Z_{A_1} \to Z_{A_1} - \Delta_A(1)$. The entity mean intervention similarly switches the binding ID vectors in $Z_{E_0}$ and $Z_{E_1}$. Additivity predicts that performing either mean intervention will reverse the model behavior: $E_0$ will be associated with $A_1$, and $E_1$ with $A_0$.

| Test condition | Control | Attribute | Entity | Both | Attribute | Entity | Both |
|---|---|---|---|---|---|---|---|
| Querying $E_0$ | 0.99 | 0.00 | 0.00 | 0.97 | 0.98 | 0.98 | 0.97 |
| Querying $E_1$ | 1.00 | 0.03 | 0.01 | 0.99 | 1.00 | 1.00 | 1.00 |

Table 1: Left: Mean calibrated accuracies for mean interventions on four test conditions. Columns are the test conditions, and rows are queries. Right: Mean interventions with random vectors.

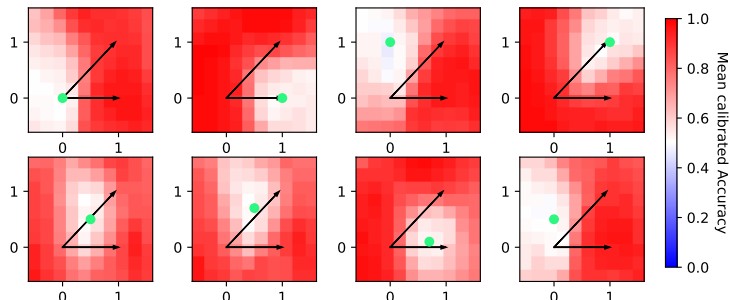

Figure 5: The plots show the mean median-calibrated accuracy when one pair of binding ID, $v_0$, is fixed at the green circle, and the other, $v_1$, is varied across the grid. The binding IDs $b(0)$, $b(1)$, and $b(2)$ are shown as the origin of the arrows, the end of the horizontal arrow, and the end of the diagonal arrow respectively. We use LLaMA-13b for computational reasons.

**Experiments.** In our experiments, we fix $n = 2$ and use 500 samples to estimate $\Delta_E(1)$ and $\Delta_A(1)$. We then perform four tests, and evaluate the model accuracy under the original belief. The Control test has no interventions, and the accuracy reflects model's base performance. The Attribute and Entity tests perform the attribute and entity mean interventions, which should lead to a complete switch in model beliefs so that the accuracy is near 0. Table 1 shows agreement with additivity: the accuracies are above 99% for Control, and below 3% for Attribute and Entity.

As a further check, we perform both attribute and entity mean interventions simultaneously, which should cancel out and thus restore accuracy. Indeed, Table 1 shows that accuracy for Both is above 97%. Finally, to show that the *specific* directions obtained by the difference vectors matter, we sample random vectors with the same magnitude but random directions, and perform the same mean interventions with the random vectors. These random vectors have no effect on the model behavior.

### 4.2 THE GEOMETRY OF BINDING ID VECTORS

§4.1 shows that we can think of binding IDs as pairs of ID vectors, and that randomly chosen vectors do not function as binding IDs. We next investigate the geometric structure of valid binding vectors and find that linear interpolations or extrapolations of binding vectors are often also valid binding vectors. This suggests that binding vectors occupy a continuous *binding subspace*. We find evidence of a metric structure in this space, such that nearby binding vectors are hard for the model to distinguish, but far-away vectors can be reliably distinguished and thus used for the binding task.

To perform our investigation, we apply variants of the mean interventions in §4.1. As before, we start with an $n = 2$ context, thus obtaining representations $Z_0 = (Z_{E_0}, Z_{A_0})$ and $Z_1 = (Z_{E_1}, Z_{A_1})$. We first erase the binding information by subtracting $(\Delta_E(1), \Delta_A(1))$ from $Z_1$, which reduces accuracy to chance. Next, we will add vectors $v_0 = (v_{E_0}, v_{A_0})$ and $v_1 = (v_{E_1}, v_{A_1})$ to the representations $Z$; if doing so restores accuracy, then we view $(v_{E_0}, v_{A_0})$ and $(v_{E_1}, v_{A_1})$ as valid binding pairs.

To generate different choices of $v$, we take linear combinations across a two-dimensional space. The basis vectors for this space are $(\Delta_E(1), \Delta_A(1))$ and $(\Delta_E(2), \Delta_A(2))$ obtained by averaging across an $n = 3$ context. Fig. 5 shows the result for several different combinations, where the coordinates of $v_0$ are fixed and shown in green while the coordinates of $v_1$ vary. When $v_1$ is close to $v_0$, the LM gets close to 50% accuracy, which indicates confusion. Far away from $v_1$, the network consistently achieves high accuracy, demonstrating that linear combinations of binding IDs (even with negative coefficients) are themselves valid binding IDs. See §G for details.

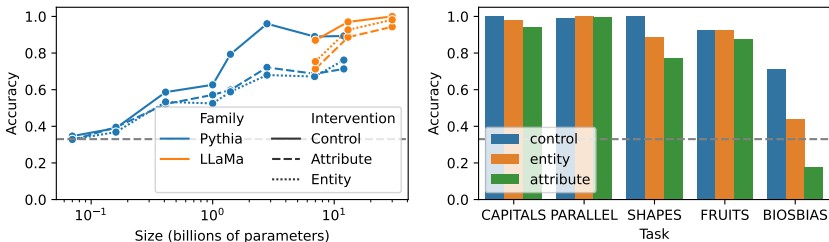

Figure 6: Left: models in Pythia and LLaMA on CAPITALS. LLaMA-65b not present for computational reasons. Right: LLaMA-30b on binding tasks. Unlike others, the BIOS task has attributes that are several tokens long.

| Task | CAPITALS | PARALLEL | SHAPES | FRUITS | BIOS | Zeros | Random |
|---|---|---|---|---|---|---|---|
| Mean accuracy | 0.88 | 0.87 | 0.71 | 0.80 | 0.47 | 0.30 | 0.31 |
| Mean log prob | -1.01 | -1.07 | -1.18 | -1.21 | -1.64 | -1.86 | -2.15 |

Table 2: The mean median-calibrated accuracy and mean log prob for mean interventions on $n = 3$ CAPITALS using binding ID estimates from other tasks. Random chance has $0.33$ mean accuracy.

The geometry of the binding subspace hints at circuits (Elhage et al., 2021) in LMs that process binding vectors. For example, we speculate that certain attention heads might be responsible for comparing binding ID vectors, since the attention mechanism computes attention scores using a quadratic form which could provide the metric over the binding subspace.

## 5    GENERALITY AND LIMITATIONS OF BINDING ID

The earlier sections investigate binding IDs for one particular task: the CAPITALS task. In this section, we evaluate their generality. We first show that binding vectors are used for a variety of tasks and models. We then show evidence that the binding vectors are task-agnostic: vectors from one task transfer across many different tasks. However, our mechanism is not fully universal. §E describes a question-answering task that uses an alternative binding mechanism.

**Generality of binding ID vectors.** We evaluate the generality of binding vectors across models and tasks. For a (model, task) pair, we compute the median-calibrated accuracy on the $n = 3$ context under three conditions: (1) the control condition in which no interventions are performed, and the (2) entity and (3) attribute conditions in which entity or attribute mean interventions (§4.1) are performed. We use the mean interventions to permute binding pairs by a cyclic shift and measure accuracy according to this shift (see §F). As shown in Figure 6, the interventions induce the expected behavior on most tasks; moreover, their effectiveness increases with model scale, suggesting that larger models have more robust structured representations.

**Transfer across tasks.** We next show that binding vectors often transfer across tasks. Without access to the binding vectors $[b_E(k), b_A(k)]$, we instead test if the difference vectors $[\Delta_E^{\text{src}}(k), \Delta_A^{\text{src}}(k)]$ from a source task, when applied to a target task, result in valid binding IDs. To do so, we follow a similar procedure to §4.2: First, we erase binding information by subtracting $[\Delta_E^{\text{tar}}(k), \Delta_A^{\text{tar}}(k)]$ from each target-task representation $[Z_{E_k}, Z_{A_k}]$, resulting in near-chance accuracy. Then, we add back in $[\Delta_E^{\text{src}}(k), \Delta_A^{\text{src}}(k)]$ computed from the *source* task with the hope of restoring performance.

Table 2 shows results for a variety of source tasks when using CAPITALS as the target task. Accuracy is consistently high, even when the target task has limited surface similarity to the target task. For example, the SHAPES task asks questions about colored shapes, and PARALLEL lists all entities before any attributes instead of interleaving them as in CAPITALS. We include two baselines for comparison: replacing $\Delta^{\text{src}}(k)$ with the zero vector ("Zeros"), or picking a randomly oriented difference vector as in Table 1 ("Random"). Both lead to chance accuracy. See §D for tasks details.

The fact that binding vectors transfer across tasks, together with the results from §4, suggests that there could be a task-agnostic subspace in the model's activations reserved for binding vectors.

## 6 RELATED WORK

**Causal mediation analysis.** In recent years, causal methods have gained popularity in post hoc interpretability (Meng et al., 2022; Geiger et al., 2021). Instead of relying on correlations, which could lead to spurious features (Hewitt & Liang, 2019), causal mediation analysis (Vig et al., 2020a) performs causal interventions on internal states of LMs to understand their causal role on LM behavior. Our work shares the same causal perspective adopted by many in this field.

**Knowledge recall.** A line of work studies recalling factual associations that LMs learn from pretraining (Geva et al., 2020; Dai et al., 2021; Meng et al., 2022; Geva et al., 2023; Hernandez et al., 2023b). This is spiritually related to binding, as entities must be associated to facts about them. However, this work studies factual relations learned from *pretraining* and how they are recalled from model *weights*. In contrast, we study representations of relations learned from *context*, and how they are recalled from model *activations*.

More recently, Hernandez et al. (2023a) found a method to construct bound representations by directly binding attribute representations to entity representations. In contrast, our work investigates bound representations constructed by the LM itself, and identifies that the binding ID mechanism (and not direct binding) is the mechanism that LM representations predominantly uses. An avenue for future work is to study how bound representations constructed by Hernandez et al. (2023a) relates to the direct binding mechanism we identified in §E.

**Symbolic representations in connectionist systems.** Many works have studied how neural networks represent symbolic concepts in activation space (Mikolov et al., 2013; Tenney et al., 2019; Belinkov & Glass, 2019; Rogers et al., 2021; Patel & Pavlick, 2021). To gain deeper insights into how these representations are used for reasoning, recent works have studied representations used for specialized reasoning tasks (Nanda et al., 2023; Li et al., 2022; 2021). Our work shares the motivation of uncovering how neural networks implement structured representations that enable reasoning.

**Mechanistic Interpretability.** Mechanistic interpretability aims to uncover circuits (Elhage et al., 2021; Wang et al., 2022; Wu et al., 2023), often composed of attention heads, embedded in language models. In our work, we study language model internals on a more coarse-grained level by identifying structures in representations that have causal influences on model behavior. Concurrent work by Prakash et al. (2024) complements ours by analyzing the circuits involved in the binding problem.

## 7 CONCLUSION

In this paper we identify and study the binding problem, a common and fundamental reasoning subproblem. We find that pretrained LMs can solve the binding task by binding entities and attributes to abstract binding IDs. Then, we identify that the binding IDs are vectors from a binding subspace with a notion of distance. Lastly, we find that the binding IDs are used broadly for a variety of binding tasks and are present in all sufficiently large models that we studied.

Taking a broader view, we see our work as a part of the endeavor to interpret LM reasoning by decomposing it into primitive skills. In this work we identified the binding skill, which is used in several settings and has a simple and robust representation structure. An interesting direction of future work would be to identify other primitive skills that support general purpose reasoning and have similarly interpretable mechanisms.

Our work also suggests that ever-larger LMs may still have interpretable representations. A common intuition is that larger models are more complex, and hence more challenging to interpret. Our work provides a counterexample: as LMs become larger, their representations can become *more* structured and interpretable, since only the larger models exhibited binding IDs (Fig. 6).

Speculating further, the fact that large enough models in two unrelated LM families learn the same structured representation strategy points to a convergence in representations with scale. Could there be an ultimate representation that these LMs are converging towards? Perhaps the properties of natural language corpora and LM inductive biases lead to certain core representation strategies that are invariant to changes in model hyperparameters or exact dataset composition. This would encouragingly imply that interpretability results can transfer across models—studying the core representations of any sufficiently large model would yield insights into other similarly large models.

ACKNOWLEDGMENTS

We thank Danny Halawi, Fred Zhang, Erik Jenner, Cassidy Laidlaw, Shawn Im, Arthur Conmy, Shivam Singhal, and Olivia Watkins for their helpful feedback. JF was supported by the Long-Term Future Fund. JS was supported by the National Science Foundation under Grants No. 2031899 and 1804794. In addition, we thank Open Philanthropy for its support of both JS and the Center for Human-Compatible AI.

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

## A  EVALUATION DETAILS

In all of our evaluations, we sample $N = 100$ instances of contexts from the binding task, obtaining $\{\mathbf{c}_i\}_{i=1}^N$. For succinctness, we write $E_k^{(i)} := E_k(\mathbf{c}_i)$ and $A_k^{(i)} := A_k(\mathbf{c}_i)$. For the $i$-th context instance, we query $E_0^{(i)}$ and $E_1^{(i)}$ which return log probabilities $\Phi_{E_0}^{(i)}(t)$ and $\Phi_{E_1}^{(i)}(t)$ over tokens $t$ in the vocabulary. However, we consider only the log probabilities for relevant attributes $\Phi_{E_k}^{(i)}(A_0^{(i)})$ and $\Phi_{E_k}^{(i)}(A_1^{(i)})$. We then compute the summary statistics (described below) over the entire population of samples so that we get two scalars, $\sigma_{E_0}$ and $\sigma_{E_1}$ describing the performance under each query entity.

- The **mean log prob** is given by $\sigma_{E_k} = \frac{1}{N}\sum_{i=1}^N \Phi_{E_k}^{(i)}(A_k^{(i)})$.
- The **top-1 accuracy** is $\sigma_{E_k} = \frac{1}{N}\sum_{i=1}^N \mathbf{1}[k = \arg\max_l \Phi_{E_k}^{(i)}(A_l^{(i)})]$.
- We adopt the **median calibrated accuracy** from Zhao et al. (2021). First, we obtain a baseline by computing medians for every attribute $m(A_l) := \mathrm{median}_{i,k}\{\Phi_{E_k}^{(i)}(A_l^{(i)})\}$. Then, compute calibrated log probs $\tilde{\Phi}_{E_k}^{(i)}(A_l) := \Phi_{E_k}^{(i)}(A_l) - m(A_l)$. The median calibrated accuracy is then $\sigma_{E_k} = \frac{1}{N}\sum_{i=1}^N \mathbf{1}[k = \arg\max_l \tilde{\Phi}^{(i)}(A_l^{(i)})]$.

Zhao et al. (2021) discusses motivations for the median calibrated accuracy. In our case, the position dependent bias provides addition reasons, which we discuss in §C.

## B  NECESSITY OF BINDING ID MECHANISM

In this section, we provide one definition of the binding ID mechanism, and argue informally that under this definition, factorizability and position independence necessarily implies the binding ID mechanism.

First, let us define the binding ID mechanism. Fix $n = 2$ for simplicity. There are two claims:

1. **Representation.** There exists a binding function $\Gamma_E$ such that for any contexts $\mathbf{c}$, $Z_{E_k}$ is represented by $\Gamma_E(E_k, k)$, in the sense that for any $e \in \mathcal{E}$, $\{Z_{E_k} \to \Gamma_E(e, k)\}$ leads to the belief $\mathbf{c}/.\{E_k \to e\}$. Likewise, there exists a binding function $\Gamma_A$ such that for any contexts $\mathbf{c}$, $Z_{A_k}$ is represented by $\Gamma_A(A_k, k)$, in the sense that for any $a \in \mathcal{A}$, $\{Z_{A_k} \to \Gamma_A(a, k)\}$ leads to the belief $\mathbf{c}/.\{A_k \to a\}$. These substitutions should also compose appropriately.

2. **Query.** Further, the binding functions $\Gamma_A$ and $\Gamma_E$ satisfy the following property: Choose any 2 permutations $\pi_E(k)$ and $\pi_A(k)$ over $\{0, 1\}$, and consider a $Z_{\text{context}}$ containing $[\Gamma_E(e_0, \pi_E(0)), \Gamma_A(a_0, \pi_A(0)), \Gamma_E(e_1, \pi_E(1)), \Gamma_A(a_1, \pi_A(1))]$. The query system will then believe $e_0 \leftrightarrow a_0, e_1 \leftrightarrow a_1$ if $\pi_E = \pi_A$, and $e_0 \leftrightarrow a_1, e_1 \leftrightarrow a_0$ otherwise.

The first claim follows from factorizability. From factorizability, we can construct the candidate binding functions simply by picking an arbitrary context consistent with the parameters. For any $e \in \mathcal{E}$ and any binding ID $k \in [0, n-1]$, pick any context $\mathbf{c}$ such that $E_k(\mathbf{c}) = e$. Then, let $\Gamma_E(e, k) = Z_{E_k}(\mathbf{c})$. $\Gamma_A$ can be constructed similarly. Our factorizability results show that the binding functions constructed this way satisfy the **Representation** claim.

The second claim follows from **Representation** and position independence. Pick an arbitrary context $\mathbf{c}$ to generate $Z_{\text{context}}$. Then, by factorizability we can make the substitutions $Z_{E_k} \to \Gamma_E(e_{\pi_E^{-1}(k)}, k)$ and $Z_{A_k} \to \Gamma_A(a_{\pi_A^{-1}(k)}, k)$, to obtain

$$[\Gamma_E(e_{\pi_E^{-1}(0)}, 0), \Gamma_A(a_{\pi_A^{-1}(0)}, 0), \Gamma_E(e_{\pi_E^{-1}(1)}, 1), \Gamma_A(a_{\pi_A^{-1}(1)}, 1)].$$

Because of factorizability, the model believes $e_0 \leftrightarrow a_0, e_1 \leftrightarrow a_1$ if $\pi_E = \pi_A$, and $e_0 \leftrightarrow a_1, e_1 \leftrightarrow a_0$ otherwise. Now, position independence lets us freely permute $\{Z_{E_0}, Z_{E_1}\}$ and $\{Z_{A_0}, Z_{A_1}\}$ without changing beliefs, which achieves the following context

$$[\Gamma_E(e_0, \pi_E(0)), \Gamma_A(a_0, \pi_A(0)), \Gamma_E(e_1, \pi_E(1)), \Gamma_A(a_1, \pi_A(1))]$$

with the desired beliefs.

## C   DETAILS FOR POSITION INDEPENDENCE

**RoPE Intervention.** In Fig. 2a, the context activations $Z_{\text{context}}$ is drawn in a line, suggesting a linear form: $Z_{\text{context}} = [\ldots, Z_{E_0}, \ldots, Z_{A_0}, \ldots, Z_{E_1}, \ldots, Z_{A_1}, \ldots]$. We can equivalently think of $Z_{\text{context}}$ as a set of pairs: $Z_{\text{context}} = \{(p, Z_p) \mid p \text{ is an index for a context token}\}$. LMs that use Rotary Position Embedding (RoPE) (Su et al., 2021), such as those in the LLaMA and Pythia families, have architectures that allow arbitrarily intervention on the apparent position of an activation $(p, Z_p) \to (p', Z_p)$, even if this results in overall context activations that cannot be written down as a list of activations. This is because position information is applied at every layer, and not injected into the residual stream like in absolute position embeddings. Specifically, equation 16 in Su et al. (2021) provides the definition of RoPE (recreated verbatim as follows):

$$q_m^{\mathsf{T}} k_n = (\mathbf{R}_{\Theta,m}^d \mathbf{W}_q x_m)^{\mathsf{T}} (\mathbf{R}_{\Theta,n}^d \mathbf{W}_k x_n) \tag{3}$$

Then, making the intervention $\mathbf{R}_{\Theta,n}^d \to \mathbf{R}_{\Theta,n^*}^d$ changes the apparent position of the activations at position $n$ to the position at $n^*$.

**Is the position dependent bias just a bias?** For the purposes of determining if *position* encodes *binding*, the fact that the LM does not substantially change its beliefs when we switch the positions of the attribute activations (or the entity activations) suggests that position can only play a limited role. However, calling the position dependency of attributes a "bias" implies that it is an artifact that we should correct for. To what extent is this true?

The case for regarding it as a bias is two-fold. First, as discussed by Su et al. (2021), RoPE exhibits long-term position decay, which systematically lowers the attention paid to activations that are further away from the query (i.e. earlier in the context). Plausibly, at some point when computing the query mechanism, the LM has to make a decision whether to pay attention to the first or the second attribute, and the presence of the long-term position decay can bias this decision, leading to the position dependent bias in the final prediction that we see.

The second reason is that there are systematic and unbiased ways of calibrating the LM to recover the correct answer, in spite of the position dependent bias. We discuss two strategies. Because the position dependent bias modifies the log probs for $A_0$ (or $A_1$) regardless of which entity is being queried, we can estimate this effect by averaging the log probs for $A_0$ (or $A_1$) for both queries $E_0$ and $E_1$. Then, when making a prediction, we can subtract this average from the log probs for $A_0$ (or $A_1$). This corresponds to the *median calibrated accuracy* metric discussed earlier. The second procedure to mitigate the position dependent bias is an intervention to set all attribute activations to have the same position, which limits the amount of bias position dependency can introduce.

These procedures do not require foreknowledge of what the ground truth predicates are, and hence do not leak knowledge into the prediction process — if the calibrated LM answers queries correctly, the information must have come from the context activations and not from the calibration process.

Nonetheless, there are features about the position dependent bias that could be interesting to study. For example, we might hope to predict the magnitudes of the position dependent bias based on RoPE's parameters. However, such an investigation will most likely involve a deeper mechanistic understanding of the query system, which we leave as future work.

# D BINDING TASK DETAILS

## D.1 CAPITALS

Construct a list of one-token names and a list of country-capital pairs that are also each one-token wide. Then, apply the following template:

```
Answer the question based on the context below. Keep the answer short.

Context: {E_0} lives in the capital city of {A_0}.
{E_1} lives in the capital city of {A_1}.

Question: Which city does {qn_subject} live in?

Answer: {qn_subject} lives in the city of
```

The LM is expected to answer with the capital of the country that is bound to the queried entity. Note that the LM is expected to simultaneously solve the factual recall task of looking up the capital city of a country.

## D.2 PARALLEL

The PARALLEL task uses the same country capital setup, but with the prompt template:

```
Answer the question based on the context below. Keep the answer short.

Context: {E_0} and {E_1} live in the capital cities of {A_0} and {A_1}
    respectively.

Question: Which city does {qn_subject} live in?

Answer: {qn_subject} lives in the city of
```

This prompt format breaks the confounder in the CAPITALS task that entity always appear in the same sentence as attributes, suggesting binding ID is not merely a syntactic property.

## D.3 FRUITS

The FRUITS task uses the same set of names, but for attributes it uses a set of common fruits and food that are one-token wide. The prompt format is:

```
Answer the question based on the context below. Keep the answer short.

Context: {E_0} likes eating the {A_0}. {E_1} likes eating the {A_1}
    respectively.

Question: What food does {qn_subject} like?

Answer: {qn_subject} likes the
```

## D.4 SHAPES

The SHAPES tasks have entities which are one-token wide *colors*, and attributes which are one-token wide *shapes*. The prompt looks like:

```
Answer the question based on the context below. Keep the answer short.
```

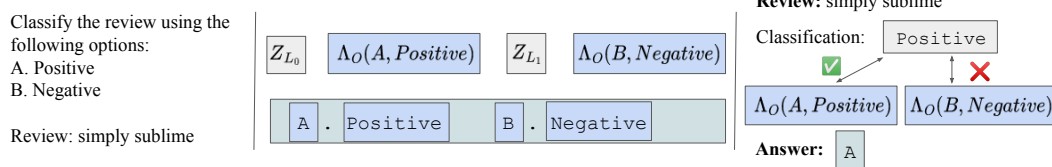

Figure 7: Direct binding in MCQ task. $O_k$ and $L_k$ denote options and labels respectively. Under direct binding, $Z_{O_0}$ and $Z_{O_1}$ are represented by a binding function $\Lambda_O$ that directly binds option and label together, whereas $Z_{L_0}$ and $Z_{L_1}$ are causally irrelevant.

```
Context: The {A_0} is {E_0}. The {A_1} is {E_1}.

Question: Which shape is colored {qn_subject}?

Answer: The {qn_subject} shape is
```

This task inverts the assumption that entities have to be nouns, and attributes are adjectives.

## D.5   BIOS

This task is adapted from the bias in bios dataset De-Arteaga et al. (2019), with a prompt format following Hernandez et al. (2023a). The entities are the set of one-token names, and the attributes are a set of biography descriptions obtained using the procedure from Hernandez et al. (2023a). The LM is expected to infer the occupation from this description. This time, the attributes are typically one sentence long, and are no longer one-token wide. We thus do not expect the mean interventions for attributes to work, although we may still expect entity interventions to work. Just inferring the correct occupation is also a much more challenging task than the other synthetic tasks.

The prompt format is:

```
Answer the question based on the context below. Keep the answer short.

Context:
About {E_0}: {A_0}
About {E_1}: {A_1}

Question: What occupation does {qn_subject} have?
Answer: {qn_subject} has the occupation of
```

## E   MCQ TASK

**Direct binding in MCQ.** While binding IDs are used for many tasks, they are not universal. We briefly identify an alternate binding mechanism, the *direct binding* mechanism, that is used for a multiple-choice question-answering task (MCQ). In MCQ, each label (A or B) has to be bound to its associated option text. In this task, instead of binding variables to an abstract binding ID, the model directly binds the label to the option (Fig. 7).

Multiple choice questions (MCQs) can be formulated as a binding task if we put the options *before* the question. This is to force the LM to represent the binding between label and option text before it sees the questions. We study the SST-2 task (Socher et al., 2013), which is a binary sentiment classification task on movie reviews (either positive or negative). Then, the attributes are single letter labels from A to E, and the entities are "Positive" and "Negative".

The prompt is as follows:

```
Classify the review using the following options:
{A_0}: {E_0}
```

```
{A_1}: {E_1}
Review: {question}
Answer:
```

Then, when prompted with a question with a certain sentiment, the LM is expected to retrieve its corresponding label.

### E.1 EXPERIMENTS

It turns out that the reversed MCQ format is too out of distribution for LLaMA-30b to solve. However, we find that the instruction finetuned tulu-13b model (Wang et al., 2023) is able to solve this task.

We find that the activations for this task are not factorizable in the same way. Consider the target context:

```
C: Negative
A: Positive
```

and the source context:

```
A: Negative
C: Positive
```

We denote the labels as $L_0$ and $L_1$, so that $L_0$ is A in the first context and B in the second context. We denote the option texts as $O_0$ and $O_1$.

We perform an experiment where we intervene by copying over a suffix of every line from the source context into the target context, and plot the accuracy based on whether the intervention successfully changes the belief (Fig. 8). The right most point of the plot is the control condition where no interventions are made. The accuracy is near zero because the model currently believes in the original context. At the left most point, we intervene on the entire statement, which is a substitution of the entire $Z_{\text{context}}$. Thus, we observe a near perfect accuracy.

Interestingly, copying over the activations for the tokens corresponding to "ative" and the whitespace following it suffices for almost completely changing the belief, despite having a surface token form that is identical at those two tokens ("ative $\langle WS \rangle$" for both source and target contexts). This suggests that those activations captures the binding information that contains both the label and the option text. This leads to the conclusion that binding information is bound directly at those activations, instead of indirectly via binding IDs.

In contrast, binding ID would have predicted that substituting these two tokens would not have made a difference, because the option activations $Z_O$ should contain only information about the option text and the binding ID, which is identical for our choice of source and target contexts.

## F  GENERALITY DETAILS

Suppose $\pi$ is a cyclic shift, say $\pi(0) = 1, \pi(1) = 2, \pi(2) = 0$. Then, we can perform mean interventions based on the cyclic shift on entities as follows:

$$Z_{E_k} \to Z_{E_k} + b_E(\pi(k)) - b_E(k) = Z_{E_k} + \Delta_E(\pi(k)) - \Delta_E(k).$$

We then expect the belief to follow the same shift, so that the LM believes $E_k \leftrightarrow A_{\pi(k)}$.

Similarly, we can perform mean interventions on attributes as follows:

$$Z_{A_k} \to Z_{A_k} + b_A(\pi(k)) - b_A(k) = Z_{A_k} + \Delta_A(\pi(k)) - \Delta_A(k).$$

However, this time we expect the belief to follow the inverse shift, i.e. $E_k \leftrightarrow A_{\pi^{-1}(k)}$, which is the same as $E_{\pi(k)} \leftrightarrow A_k$.

As usual, we sample $\Delta$ using 500 samples. We perform the intervention using both cyclic shifts over 3 elements, (i.e. $\pi$ and $\pi^{-1}$), and report the mean results over these two shifts.

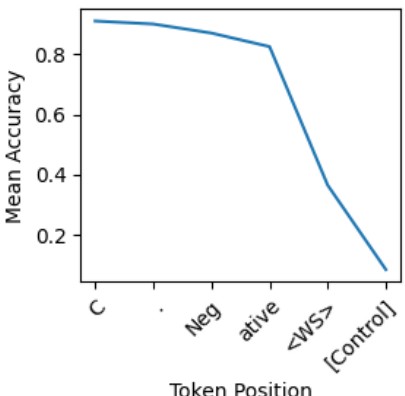

Figure 8: Substitutions for MCQ option suffix

# G  GEOMETRY DETAILS

An experimental challenge we face is that we do not have access to the binding ID vectors $b_A, b_E$ themselves, only differences between them, $\Delta_A, \Delta_E$. For clarity of exposition we first describe the procedure we would perform if we had access to the binding ID vectors, before describing the actual experiment.

In the ideal case, we would obtain two pairs of binding ID vectors, $[b_E(0), b_A(0)], [b_E(1), b_A(1)]$. Then, we can construct two linear combinations of these two binding ID vectors as candidate binding IDs, $[v_{E_0}, v_{A_0}]$ and $[v_{E_1}, v_{A_1}]$. Now, we can take an $n = 2$ context $\mathbf{c}$ and intervene on each of $Z_{E_0}, Z_{A_0}, Z_{E_1}, Z_{A_1}$ to change their binding IDs to our candidate binding IDs. If the model retains its beliefs, then we infer that the binding IDs are valid.

There two main problems with this procedure. The first is that we only have access to $\Delta_A$ and $\Delta_E$ and not $b_E, b_A$. Instead of choosing $[b_E(0), b_A(0], [b_E(1), b_A(1)]$ as our basis vectors, we can use contexts with $n = 3$ to obtain $[\Delta_E(1), \Delta_A(1)], [\Delta_E(2), \Delta_A(2)]$. These new basis vectors are still linear combinations of binding IDs, and if binding ID vectors do form a subspace, these would be part of the subspace too.

The second problem is that we cannot arbitrarily set the binding ID vector of an activation to another binding ID vector. Instead, we can only add vectors to activations. We thus perform two sets of interventions. We first perform the mean interventions on the second binding ID pair to turn $[b_E(1), b_A(1)]$ into $[b_E(0), b_A(0)]$. At this point, the LM sees two entities with the same binding ID and two attributes with the same binding ID, and is confused. Then, we can add candidate binding vector ID *offsets* to these activations.

More precisely, let $\eta, \nu$ be coefficients for the linear combinations of the basis vectors. Define now $h_A(\eta, \nu) = \eta\Delta_A(1) + \nu\Delta_A(2)$ and $h_E(\eta, \nu) = \eta\Delta_E(1) + \nu\Delta_E(2)$ as the candidate binding vector ID offsets. Then, we add $[h_E(\eta_0, \nu_0), h_A(\eta_0, \nu_0)]$ and $[h_E(\eta_1, \nu_1), h_A(\eta_1, \nu_1)]$ to the respective two pairs of binding IDs, and evaluate if the model has regained its beliefs.

Concretely, the intervention we apply is parameterized by $(\eta_0, \nu_0, \eta_1, \nu_1)$ and are as follows:

$$Z_{A_0} \to Z_{A_0} - \Delta_A(0) + h_A(\eta_0, \nu_0), \quad Z_{E_0} \to Z_{E_0} - \Delta_E(0) + h_E(\eta_0, \nu_0),$$

$$Z_{A_1} \to Z_{A_1} - \Delta_A(1) + h_A(\eta_1, \nu_1), \quad Z_{E_1} \to Z_{E_1} - \Delta_E(1) + h_E(\eta_1, \nu_1).$$

We are now interested in the question: if we have coefficients $(\eta_0, \nu_0)$ and $(\eta_1, \nu_1)$, are the binding vectors constructed from those coefficients valid binding IDs?

In our experiments (Fig. 5), we fix the value of $\eta_0$ and $\nu_0$ at varying positions (green circles), and vary $\eta_1$ and $\nu_1$. We plot the mean median-calibrated accuracy. We find that near the green circle, the model is completely confused, responding with near-chance accuracy. This verifies that the erasure step works as intended. In addition, we find that there appears to be a binding metric subspace in that as long as candidate binding IDs are sufficiently far apart, the LM recovers its ability to distinguish

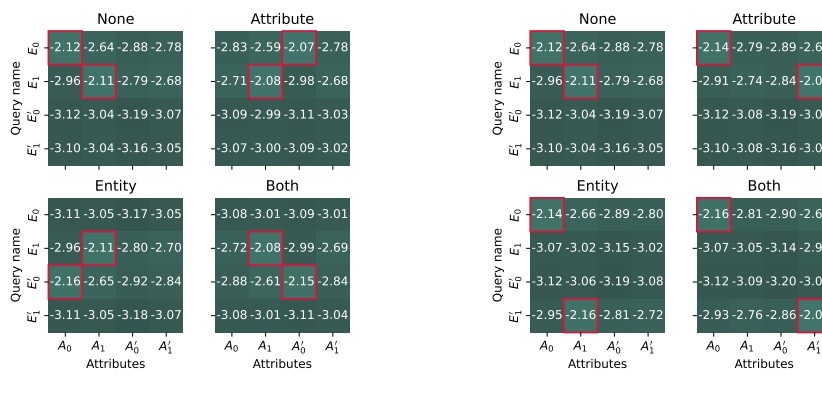

(a) Swapping entity/attribute for $(E_0, A_0)$      (b) Swapping entity/attribute for $(E_1, A_1)$

Figure 9: Factorizability results for ONEHOP

| Test condition | Control | Attribute | Entity | Both |
|---|---|---|---|---|
| Querying $E_0$ | 0.73 | 0.25 | 0.24 | 0.71 |
| Querying $E_1$ | 0.79 | 0.28 | 0.26 | 0.77 |

Table 3: Mean intervention results for ONEHOP

between the two, even when candidate binding IDs are outside of the convex hull between the three binding IDs used to generate the basis vectors.

# H  ONE HOP EXPERIMENT

One sign that the binding ID mechanism correctly captures the semantic binding information is that the LM is able to reason with representations modified according to the binding ID theory.

To some extent, the CAPITALS task already requires a small reasoning step: in the context the LM is given that "Alice lives in the capital city of France", and is asked to answer "Paris". This means that binding mechanism that binds "Alice" to "France" has to create representations that are robust enough to support the inference step "France" to "Paris".

To further push on reasoning, we introduce the ONEHOP task, an augmented version of CAPITALS. The context remains the same as the CAPITALS task, i.e. we provide a list of people and where they live. However, the LM has to apply an additional reasoning step to answer the question. An example context and question is below:

```
Answer the question based on the context below. Keep the answer short.

Context: Elizabeth lives in the capital city of France. Ryan lives in the
    capital city of Scotland.

Question: The person living in Vienna likes rose. The person living in
    Edinburgh likes rust. The person living in Tokyo likes orange. The
    person living in Paris likes tan. What color does Elizabeth like?

Answer: Elizabeth likes
```

In the ONEHOP task, based on the binding information in the context the LM has to perform two inference steps. The first is to infer that the capital city of France is Paris. The second is to, based on the additional information in the question, infer that the person living in Paris likes tan, and output tan as the correct answer.

This is a more challenging task than our other tasks, and we thus present results on LLaMA-65b instead of LLaMA-30b. Overall, we find that all of our results still hold. We show results for factorizability (Fig. 9), position independence (Fig. 10), and mean interventions (Fig. 3).

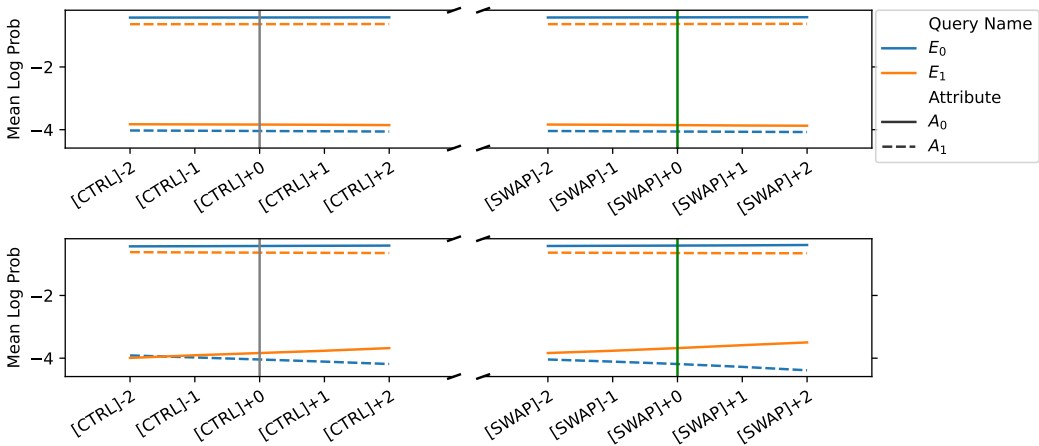

Figure 10: Position independence for ONEHOP. Top: Mean log probs for entity interventions. Bottom: Mean log probs for attributes. Different from Fig. 4, we only compute the local neighborhood around the control and swapped conditions.

## I   THREE-TERM BINDING

In all of our tasks, we studied binding between two terms: binding an entity to an attribute. Here, we extend our results to three-term binding. An example context looks like:

```
Answer the question based on the context below. Keep the answer short.

Context: Carol from Italy likes arts. Samuel from Italy likes swimming.
    Carol from Japan likes hunting. Samuel from Japan likes sketching.

Question: What does Carol from Italy like?

Answer: Carol from Italy likes
```

In general, each statement in the context binds three terms together: a name, a country, and a hobby. We can query any two of the three terms, and ask the language model to retrieve the third. The above example shows how we query for the hobby, given the name and the country. We query for country and name by asking instead:

```
Which country is Carol who likes hunting from?
```

```
Who from Italy likes hunting?
```

We extend our analysis to three-term binding in the following way. Of the three attribute classes, namely names, countries, and hobbies, choose one to be the *fixed* attribute, one to be the *query* attribute, and one to be the *answer* attribute. Altogether, there are $3! = 6$ possible assignments. For every such assignment, we can perform the same set of analysis as before.

To illustrate, suppose we choose country as the fixed attribute, name to be the query attribute, and hobby to be the answer attribute. An example prompt for this assignment will look like:

```
Answer the question based on the context below. Keep the answer short.

Context: Carol from Italy likes arts. Samuel from Italy likes swimming.

Question: What does Carol from Italy like?

Answer: Carol from Italy likes
```

| Fixed | Query | Answer | Test condition | Control | Attribute | Entity | Both |
|---|---|---|---|---|---|---|---|
| name | country | hobby | Query 0 | 1.00 | 0.00 | 0.00 | 1.00 |
| name | country | hobby | Query 1 | 1.00 | 0.00 | 0.00 | 1.00 |
| name | hobby | country | Query 0 | 1.00 | 0.00 | 0.00 | 1.00 |
| name | hobby | country | Query 1 | 1.00 | 0.00 | 0.00 | 1.00 |
| country | name | hobby | Query 0 | 1.00 | 0.01 | 0.00 | 1.00 |
| country | name | hobby | Query 1 | 1.00 | 0.00 | 0.00 | 1.00 |
| country | hobby | name | Query 0 | 1.00 | 0.02 | 0.01 | 0.99 |
| country | hobby | name | Query 1 | 1.00 | 0.03 | 0.01 | 0.99 |
| hobby | name | country | Query 0 | 1.00 | 0.00 | 0.00 | 1.00 |
| hobby | name | country | Query 1 | 1.00 | 0.00 | 0.00 | 1.00 |
| hobby | country | name | Query 0 | 1.00 | 0.00 | 0.00 | 1.00 |
| hobby | country | name | Query 1 | 1.00 | 0.00 | 0.00 | 1.00 |

Table 4: Mean intervention results for three-term binding. The intervened model perform near perfectly for most test conditions.

We then report the median-calibrated accuracy for the mean interventions under all 6 assignments (Table 4). The accuracy is better than CAPITALS (Table 1) because CAPITALS requires inferring capital city from country, whereas ONEHOP only requires looking up and copying countries.

## J ADDITIONAL FACTORIZABILITY AND POSITION INDEPENDENCE PLOTS

This section contains the experiments for factorizability (Fig. 3) and position independence (Fig. 4) reproduced for the other binding tasks, namely PARALLEL (Fig. 11, 15), SHAPES (Fig. 12, 16), FRUITS (Fig. 13, and BIOS (Fig. 14).

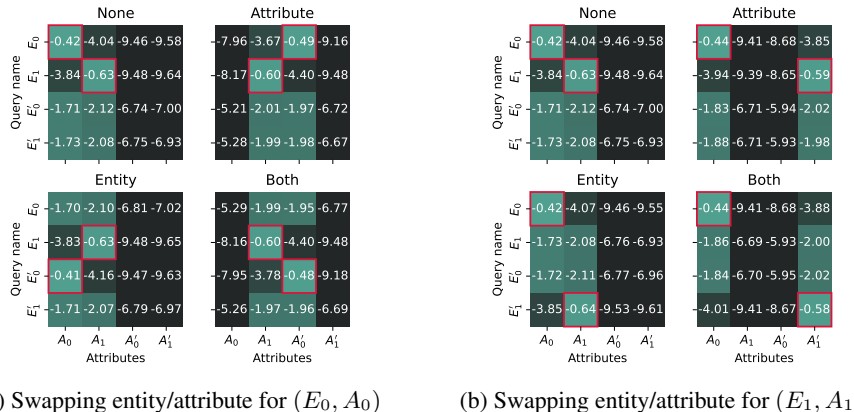

(a) Swapping entity/attribute for $(E_0, A_0)$  (b) Swapping entity/attribute for $(E_1, A_1)$

Figure 11: Factorizability results for PARALLEL

Notice that for BIOS (Fig. 14), entity factorizability works, but not attribute factorizability. This is because the attribute information is represented by many tokens, while the attribute factorizability test only substitutes the first token in the attribute representation.

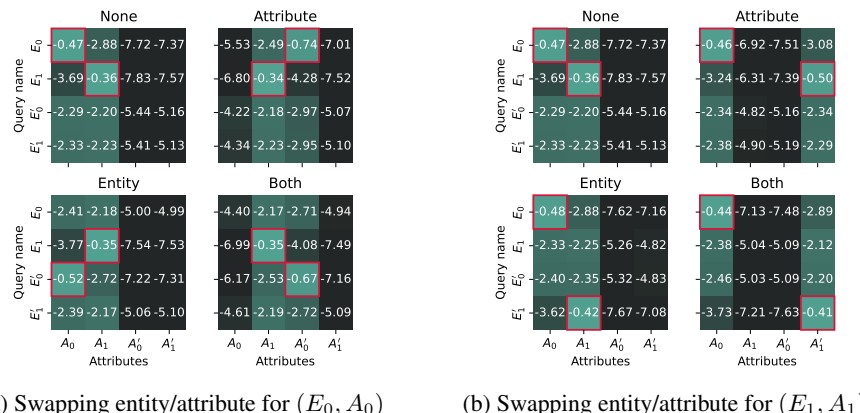

(a) Swapping entity/attribute for $(E_0, A_0)$     (b) Swapping entity/attribute for $(E_1, A_1)$

Figure 12: Factorizability results for SHAPES

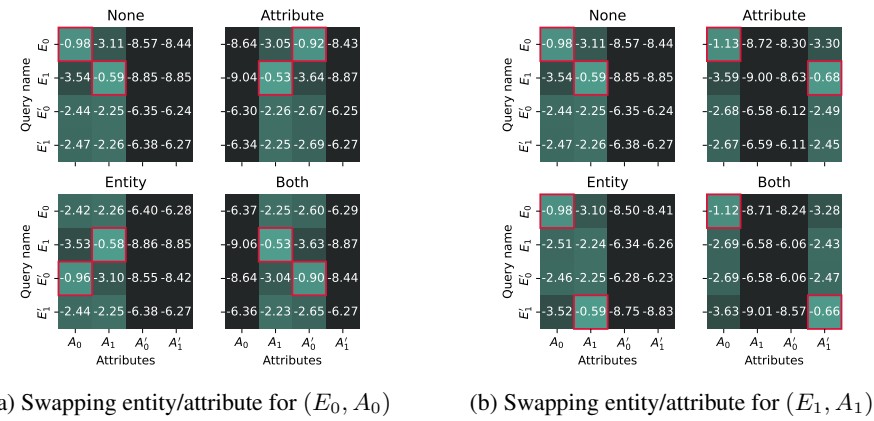

(a) Swapping entity/attribute for $(E_0, A_0)$     (b) Swapping entity/attribute for $(E_1, A_1)$

Figure 13: Factorizability results for FRUITS

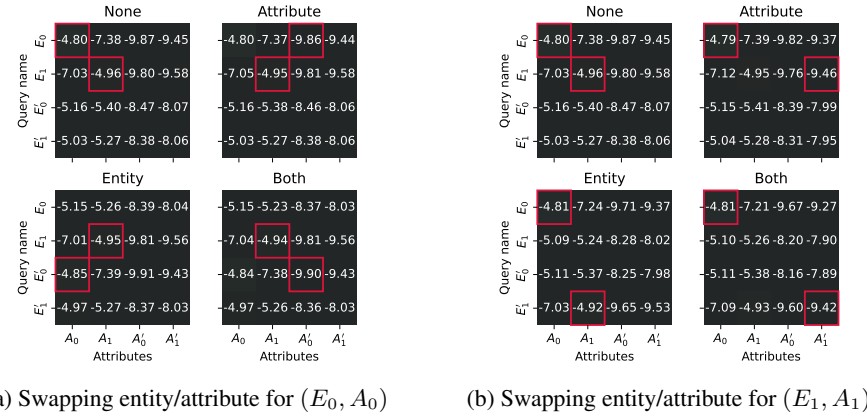

(a) Swapping entity/attribute for $(E_0, A_0)$     (b) Swapping entity/attribute for $(E_1, A_1)$

Figure 14: Factorizability results for BIOS

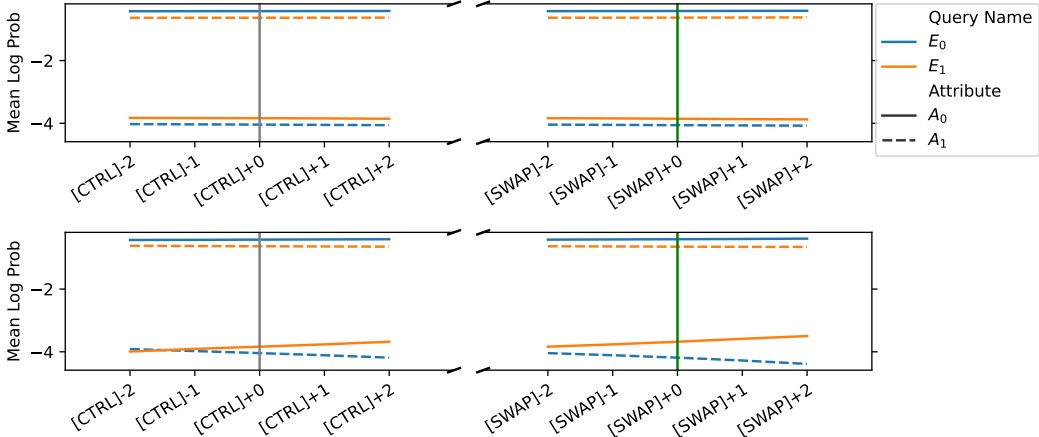

Figure 15: Position independence for PARALLEL. Top: Mean log probs for entity interventions. Bottom: Mean log probs for attributes. Different from Fig. 4, we only compute the local neighborhood around the control and swapped conditions.

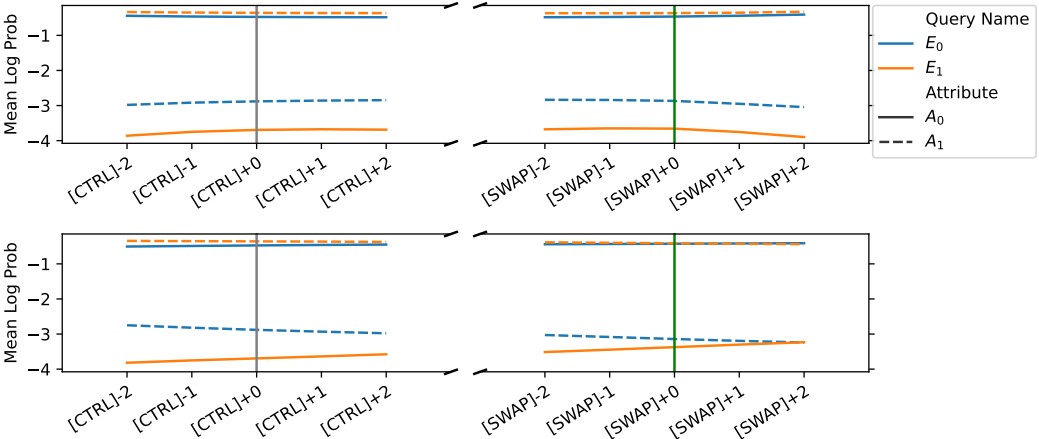

Figure 16: Position independence for SHAPES. Top: Mean log probs for entity interventions. Bottom: Mean log probs for attributes. Different from Fig. 4, we only compute the local neighborhood around the control and swapped conditions.

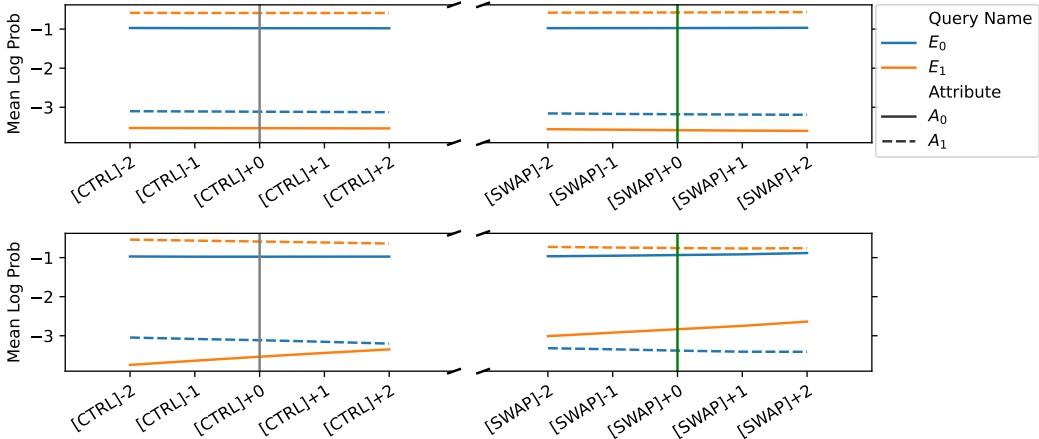

Figure 17: Position independence for FRUITS. Top: Mean log probs for entity interventions. Bottom: Mean log probs for attributes. Different from Fig. 4, we only compute the local neighborhood around the control and swapped conditions.

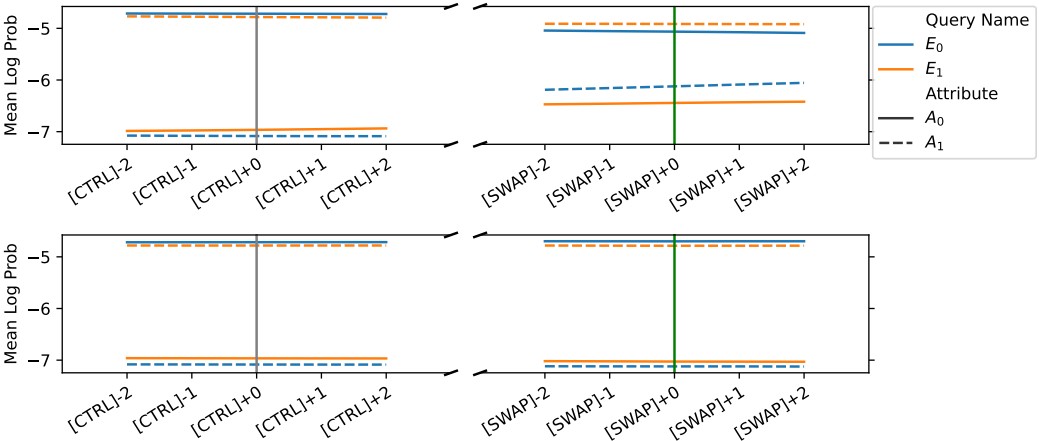

Figure 18: Position independence for BIOS. Top: Mean log probs for entity interventions. Bottom: Mean log probs for attributes. Different from Fig. 4, we only compute the local neighborhood around the control and swapped conditions.

