# OpenReview forum: "How do Language Models Bind Entities in Context?"
_ICLR.cc/2024/Conference — ICLR 2024 poster_

### Official Review · Reviewer_aQw7 · 2023-10-26

**Soundness:** 3 good
**Presentation:** 2 fair
**Contribution:** 3 good
**Rating:** 5
**Confidence:** 2

**Summary:**

This paper investigates a binding problem in which language models must represent the associations among entities and attributes. The authors empirically check the behavior of binding considering factorizability and position independence, and they introduce the idea of a binding ID vector, considering the additivity of binding functions. The paper shows several pieces of evidence that imply the existence of the binding ID vectors and their generalities.

**Strengths:**

- The paper proposes novel binding ID vectors that can help understand LLM's behavior.
- The paper presents empirical evidence of binding by checking substitution and position independence.
- Analyses are provided for biding ID vectors, including geometry and generalization.

**Weaknesses:**

- The paper addresses a simple binding problem between a pair of entities. It is unclear how this can be generalized to the binding behavior on more than binary relations and entities involved in multiple binding.
- Some details are not clear, as in the following questions.

**Questions:**

- When the authors substitute activations in Figure 3 or calculate median-calibrated accuracy in Figure 5 and Table 2, which part of the model is substituted and which is frozen? LLMs are autoregressive models, and during decoding, when and how are the parameters substituted?
- Among what did the authors calculate the mean of log probs in Figure 4?
- In equation (2), they do sampling, but from what did the authors sample?
- The lines in the legend do not match those in the graph in Figure 4. In the bottom graph, the mean log probs increase or decrease with the positions, but the authors say they are small changes. How can they be said small?

**Details Of Ethics Concerns:**

No problem.

---

> ### Author Response · Authors · 2023-11-16
>
> Thank you for the comments.
>
> > When the authors substitute activations in Figure 3 or calculate median-calibrated accuracy in Figure 5 and Table 2, which part of the model is substituted and which is frozen? LLMs are autoregressive models, and during decoding, when and how are the parameters substituted?
>
> The internal activations corresponding to the context (i.e. the part of the prompt that contains the binding information) is frozen (denoted as $Z_{\text{context}}$ in the paper). We do not intervene on the _parameters_, but rather we substitute a subset of $Z_{\text{context}}$, and observe how the new context activations change the behavior of the LM. This procedure is explained in greater detail in section 2.2 of the initial submission, which, in the current version, is the second half of section 2.
>
> > Among what did the authors calculate the mean of log probs in Figure 4?
>
> In Fig. 4, we sampled 100 2-entity contexts from the CAPITALS task, with different values of entity/attributes. For each context, we perform the position interventions, and the resulting log probs are averaged across the 100 samples.
>
> > In equation (2), they do sampling, but from what did the authors sample?
>
> In general when obtaining estimates for $\Delta(k)$ we draw 500 samples of different contexts (with each entity/attribute chosen uniformly at random without repetition), but with the same number of entity/attribute pairs.
>
> > The lines in the legend do not match those in the graph in Figure 4. In the bottom graph, the mean log probs increase or decrease with the positions, but the authors say they are small changes. How can they be said small?
>
> The changes in log prob are small in that they do not change the prediction of the language model. It may have become less confident in its predictions, but it still outputs the correct answer.
>
> In addition, we interpret the overall trend in the mean log probs as a _position dependent bias_. This is is a bias because regardless of which entity is being queried, shifting an attribute to a later position always makes it have a higher log probability. We speculate that this is due to the long-term decay property of [Rotary Position Embedding (RoPE)](https://arxiv.org/abs/2104.09864) that the original authors discussed.
>
> We account for this bias using the median-calibrated accuracy instead of accuracy in our experiments. In appendix C we discussion this bias further, and we have also updated the main text to contain some of this reasoning.

---

> ### Author Response · Authors · 2023-11-21
>
> > It is unclear how this can be generalized to the binding behavior on more than binary relations and entities involved in multiple binding.
>
> We have conducted new experiments that test binding with three terms. Please see our [response to reviewer H1Kh](https://openreview.net/forum?id=zb3b6oKO77&noteId=BYsiAMpu7m).

---

### Official Review · Reviewer_H1Kh · 2023-10-30

**Soundness:** 3 good
**Presentation:** 2 fair
**Contribution:** 4 excellent
**Rating:** 6
**Confidence:** 2

**Summary:**

This paper proposes a hypothetical mechanism to explain how LLMs associate entities to specific properties in the in-context learning setting, which the authors call the _binding problem_. The phenomenon is mostly studied in the context of a very short and simple reasoning task, and the key investigative tool is _causal mediation analysis_, where carefully chosen sets of internal activations are substituted into to the network before generating responses to queries. The experimental results are consistent with a hypothesized "binding ID". Further experiments test other properties and behaviors of this mechanism.

**Strengths:**

Figure 2 was very helpful in illustrating the substitution scheme.

I am not completely convinced that the claims follow from the observations due to some clarity (or confusion on my part) issues (see below), but assuming they hold: the _binding ID_ mechanism and its properties represent a really exciting discovery in terms of understanding LLM ICL phenomena. I suppose maybe this sort of lines up with the recent (Bricken et al 2023)[https://transformer-circuits.pub/2023/monosemantic-features/index.html] and associated previous superposition research. In any case, this mechanism would unlock a lot of promising research directions as well as practical applications.

Section 3.1: the hypothesized mechanism and its consequences are, for the most part, laid out clearly and precisely. (nit: "the only information ... that affects the query behavior" _for our specific queries_)

To the extent that I correctly understood the experiments, the ability to, rather surgically, manipulate the "beliefs" of the model was fairly clear evidence consistent with the hypothesized mechanism. The setting where switching both entity and attributing binding IDs restores correct query answering was a "wow" moment for me.

**Weaknesses:**

Section 2.2: it was not immediately obvious to me whether the stacked activations completely "d-separates" (maybe not exactly this concept?) the intervention token from everything else, without some more detail on the LM architecture.

Section 4.1 is very dense, and I found it difficult to follow without working it out myself on separate paper. Given the importance of these concepts to the rest of the paper, a diagram might help make it clearer. See questions below, but I had a central confusion about some core mechanism here even after working through this pretty closely. Without resolving the clarity a bit, it's difficult to fully accept the interpretations of the experiments, as significant as the findings would be.

Section 5 supports the generality of the mechanism across other similar tasks, which further strengthens the central contribution. The existence of the alternative _direct binding_ mechanism was interesting to see and another good contribution of the paper, although it was not super clear without reading the corresponding Appendix.

**Questions:**

"If there no context consistent...say that the LM is _confused_" - I didn't fully understand this why or how this would happen, and also it didn't seem to come up elsewhere in the paper. Is it necessary to include?

"A subtle point is that ..." this seems centrally important but only mentioned in passing.

Figure 3 b seems like the wrong plot? It doesn't show what the text claims but rather seems to be a copy of Figure 3 a.

Section 4.1: what are we sampling here: from different "X lives in Y"-style context sentences? I don't totally understand why / how we'd expect the $b(k)$ vectors to be consistent for a given value of $k$ across different contexts, or is that not required? *Actually, this may be my central confusion with Section 4 and beyond*:
* in Figure 1, abstract shapes (triangle, square) are used to represent the abstract binding IDs
* in Section 4.1, binding ID $k$ is represented by some pair of functions $\[b_E(k),b_A(k)\]$
* the $\Delta$ calculation scheme in Eqn 2 and the expectation sampling seem to crucially depend on, eg $b_E(k)$ being a stable/consistent function of (the seemingly arbitrary index?) $k$ across _any context_ $\bf{c}$

So, is the hypothesized mechanism that:
1. LLM, internally, have some weird analogue of LISP `gensym` that generates arbitrary but unique abstract binding IDs like "square" and "triangle" (but of course are really lie in a linear subspace of vectors) when an entity/attribute is seen in-context?
2. OR that LLM have some stable/consistent binding function $b_E(k)$, and the first time `llm-gensym` gets called, the result for $k=0$ is returned, and some internal entity counter (?) executes $k++$ ?
3. OR some other process else altogether that I am missing?

Figure 1 led me to believe case 1, but then Section 4.1 (as far as I can tell) requires case 2, and in fact the experimental results seem consistent with that mechanism. This is even more surprising to me, and I would argue needs to be laid out a bit more explicitly, if this is indeed the correct interpretation. Or possibly, it is case 1 and I am mistaken that the rest of the work depends on the "stable $b(k)$" described in case 2?

Eqn 2: is there any significance to the notation change of introducing $\alpha$ ?

"These random vectors have no effect" - I would have expected it to break things. so I guess this is even further evidence that the binding vectors are treated in a "special" way by the LLM machinery? If so, maybe this reasoning and conclusion could be emphasized.

Perhaps due to my previous confusion about how binding ID functions work with $k$, but I could not understand what Figure 5 was conveying at all.

For any of this to make sense, I guess you are using a word-level tokenizer and not something like BPE/etc? Except it seems like the direct binding experiment uses sub-word tokens? Again, for clarity it might help to lay this out explicitly.

---

> ### Author Response · Authors · 2023-11-16
>
> Thank you for the detailed comments. We are happy that you share our excitement about how the work could have deep implications for understanding “LLM ICL phenomena”, and possibly unlocking “a lot of promising research directions as well as practical applications”.  We are also encouraged that you appreciated how our experiments could “surgically” manipulate the model _beliefs_.
>
> We’ll start by addressing your “central confusion”. You asked
> > So, is the hypothesized mechanism that:
> > 1. LLM, internally, have some weird analogue of LISP gensym that generates arbitrary but unique abstract binding IDs like "square" and "triangle" (but of course are really lie in a linear subspace of vectors) when an entity/attribute is seen in-context?
> > 2. OR that LLM have some stable/consistent binding function
>
> The answer is option 2. We find the binding ID is unchanged when we change the value of any entity or attribute. i.e. if we have:
>
> > “Alice lives in Thailand. Bob lives in France.”
>
> and
>
> > “Charlie lives in Singapore. David lives in Spain.”
>
> Then, Alice and Charlie will have the same binding ID, as will Bob and David.
>
> To generalize this, for a fixed binding task, the $k$-th entity will always have the same binding ID and the $k$-th attribute will always have the same binding ID. In section 3, we arbitrarily label the binding ID that the $k$-th entity/attribute pair has as $k$, and in section 4.1, we showed that the binding ID labelled $k$ is actually represented as a pair of binding vectors $[b_E(k), b_A(k)]$.
>
> We have updated the paper to clarify this. This is a very subtle point that you have raised and we are happy to clarify any doubts you have.

---

> ### Author Response · Authors · 2023-11-16
>
> > "These random vectors have no effect" - I would have expected it to break things. so I guess this is even further evidence that the binding vectors are treated in a "special" way by the LLM machinery? If so, maybe this reasoning and conclusion could be emphasized.
>
> Yes, one interpretation is that the binding vectors point in a specialized subspace reserved for binding IDs. Thus, random vectors outside this subspace don’t affect things that much. We also conducted a sanity check by scaling up the random vectors - at a large enough scale, the random vectors do in fact break the model's ability to bind.
>
> > Perhaps due to my previous confusion about how binding ID functions work with $k$, but I could not understand what Figure 5 was conveying at all.
>
> Perhaps you could give section 4.2 another try if we have resolved your previous confusion. At any case, we’ll give an intuitive summary:
>
> In section 4.1, we used the success of mean interventions as proof that binding IDs are really just binding ID vectors, i.e. the $k$-th binding ID is represented by the pair of entity binding and attribute binding vectors $[b_E(k), b_A(k)]$.
>
> In section 4.2, where Figure 5 is from, we try to figure out what other vectors could serve as binding vectors.
>
> One hypothesis is that there might be only a discrete sequence of binding vectors that are hardcoded into the model, and so the model simply assigns binding IDs from this discrete sequence to entities/attributes it sees. This is the imagery that your `llm-gensym` analogy evokes.
>
> Another hypothesis, which is the one we find support for, is that there is a continuous region in activation space that are all valid binding vectors. The model then, using a mechanism unknown to us, assigns binding IDs from this continuous region to entities/attributes it sees, with the condition that binding IDs have to be sufficiently far apart to be distinguished from each other.
>
> To test this, we take two entity/attribute pairs and modify their binding IDs using mean interventions (i.e. adding an offset to their binding IDs). From section 4 we know that if we add a random offset it doesn’t change anything. Instead, we obtain basis vectors for offsets by taking the differences between the first three binding IDs.
>
> In Figure 5, we show that these basis vectors are consistent with the continuous region hypothesis. Within the subspace spanned by the two basis vectors, whenever the two binding vectors are close to each other, the model loses its binding ability, and when they are far apart from each other, the model regains its binding ability.
>
> > "If there no context consistent...say that the LM is confused" - I didn't fully understand this why or how this would happen, and also it didn't seem to come up elsewhere in the paper. Is it necessary to include?
>
> We meant confusion as a catch all for when model behavior doesn’t seem consistent with any belief (perhaps because of a blunt intervention that broke things).
>
> In section 4.2 in particular, we constructed an experimental setting where we deliberately use mean interventions to make the model give equal predictions to the attributes regardless of which entity is queried (i.e. the model is completely ambivalent between the two attributes). This is also an example of confusion.
>
> > Figure 3 b seems like the wrong plot? It doesn't show what the text claims but rather seems to be a copy of Figure 3 a.
>
> Yes, this was a plotting mistake. The paper has been updated.
>
> > Eqn 2: is there any significance to the notation change of introducing $\alpha$?
>
> Yes, this is a typo.
>
> > For any of this to make sense, I guess you are using a word-level tokenizer and not something like BPE/etc? Except it seems like the direct binding experiment uses sub-word tokens? Again, for clarity it might help to lay this out explicitly.
>
> We are using sub-word tokens. In most of our tasks, the entities and attributes are chosen to be a single token wide, since a lot of common words are encoded as a single token.

---

> > ### Comment · Reviewer_H1Kh · 2023-11-18
> > **Author comments**
> >
> > Thank you for the clarifications. I believe these would strengthen the submission, especially around the key "$k$-th entity will always have the same binding ID and the $k$-th attribute will always have the same binding ID" aspect. I will increase my recommendation score accordingly.
> >
> > That said, it this is still a pretty dense / challenging read. But on the other hand, I think this work would have good potential to be influential on follow up research on in-context binding behaviors: what about > 2 attributes, alternate grammatical structures, etc?

---

> > > ### Author Response · Authors · 2023-11-21
> > >
> > > We agree that this work has identified a new theory that explains emergent in-context binding behaviors - we believe that this will be an active and fruitful line of research.
> > >
> > > > alternate grammatical structures
> > >
> > > The PARALLEL task in our original submission explores this. An example prompt from it looks like:
> > >
> > > ```
> > > Answer the question based on the context below. Keep the answer short.
> > >
> > > Context: Alice and Bob live in the capital cities of France and Thailand respectively.
> > >
> > > Question: Which city does Bob live in?
> > >
> > > Answer: Bob lives in the city of
> > > ```
> > >
> > > All of our binding experiments work for PARALLEL tasks. This shows that the binding ID is unlikely to be assigned based on simple heuristics like proximity or whether the terms are part of the same sentence, but rather involves some amount of lexical parsing.
> > >
> > >
> > > > what about > 2 attributes
> > >
> > > We have performed an additional experiment where we bind 3 terms together.
> > >
> > > ```
> > > Answer the question based on the context below. Keep the answer short.
> > >
> > > Context: Carol from Italy likes arts. Samuel from Italy likes swimming. Carol from Japan likes hunting. Samuel from Japan likes sketching.
> > >
> > > Question: What does Carol from Italy like?
> > >
> > > Answer: Carol from Italy likes
> > > ```
> > >
> > > In this binding task, there are three terms that have to be bound together: name, country, and hobby. We can query any two of the three terms, and ask the language model to retrieve the third. The above example shows how we query for the hobby, given the name and the country. We query for country and name by asking instead:
> > >
> > > To systematically study three-term binding, of the three attributes we choose one to be the _fixed_ attribute, one to be the _query_ attribute, and one to be the _answer_ attribute. Altogether, there are $3!=6$ possible assignments. For every such assignment, we can perform the same set of analysis as before.
> > >
> > > To illustrate, suppose we choose country as the fixed attribute, name to be the query attribute, and hobby to be the answer attribute. An example prompt for this assignment will look like:
> > >
> > > ```
> > > Answer the question based on the context below. Keep the answer short.
> > >
> > > Context: Carol from Italy likes arts. Samuel from Italy likes swimming.
> > >
> > > Question: What does Carol from Italy like?
> > >
> > > Answer: Carol from Italy likes
> > > ```
> > >
> > > We can then vary the query attribute (Carol or Samuel) and check if the LM returns the corresponding answer attribute (arts or swimming), analogous to how in 2-term binding we vary the entity and check if the LM returns the corresponding attribute. To test the binding ID mechanism, we apply mean interventions and verified that the accuracies flip completely when we apply mean interventions on either query or answer attributes, but not both.
> > >
> > > The full results, where the mean interventions are performed for all 6 assignments, are shown in the new Appendix I.

---

### Official Review · Reviewer_n1nh · 2023-11-01

**Soundness:** 3 good
**Presentation:** 1 poor
**Contribution:** 2 fair
**Rating:** 5
**Confidence:** 3

**Summary:**

This paper analyzes how a language model (LM) correctly associates entities and their attributes described in context (i.e. as a prompt). The authors introduce the idea of binding ID vectors and show that an LM learns a pair of binding ID vectors for each entity-attribute pair so that the LM uses the vectors to correctly identify the right attribute when answering a query about an entity. They also present experimental results suggesting that the binding vectors reside in a linear subspace and are transferable across different tasks.

**Strengths:**

- The paper explores an interesting topic on representation learning that should contribute to a deeper understanding of LLMs.
- The paper presents a novel idea to explain the phenomenon.and experimental results that support the idea.

**Weaknesses:**

- The paper is sometimes difficult to follow. This may be because the main body of the paper contains too many concepts and fails to provide important explanations and examples that would help the reader understand the concepts.
- It is not entirely clear whether the authors’ conclusions are supported by experimental evidence.

**Questions:**

- Section 2.1: What exactly do you mean by “the k-th entity” and “the k-th attribute”? Is the entity that appears first in the sentence called the 0-th entity?
- Section 2.1: What did you sample N=100 contexts from?
- Section 3.2: What is the experimental setting in the experiment?
- Section 4.2: Does the experimental result really mean that the vectors that are not linear combinations of binding ID vectors do not work?
- Figure 5: What are the x and y axes? Are they the coefficients of the basis vectors?
- p. 8: What exactly is a cyclic shift?
- p. 2: Does a “sample” mean an example?  In statistics, a sample usually means a collection of examples (data points).
- p. 1: alternate -> alternative?
- p. 2: of of -> of
- p. 8 In See -> See

---

> ### Author Response · Authors · 2023-11-16
>
> Thanks for the feedback!
>
> > Section 2.1: What exactly do you mean by “the k-th entity” and “the k-th attribute”? Is the entity that appears first in the sentence called the 0-th entity?
>
> Yes, we use 0-indexing. This is also consistent with the notation of referring to the entities in an $n$-entity context as $E_0, E_1, \dots, E_{n-1}$ and the attributes as $A_0, \dots, A_{n-1}$.
>
> > Section 2.1: What did you sample N=100 contexts from?
>
> Given a binding task with a set of possible entities $\mathcal E$ and attributes $\mathcal A$ (e.g. $\mathcal E$ is a set of first names and $\mathcal A$ is a set of countries in the CAPITALS task), we can sample an $n$-entity context by drawing $n$ entities uniformly at random without repetition from $\mathcal E$, and $n$ attributes uniformly at random without repetition from $\mathcal A$.
>
> > Section 3.2: What is the experimental setting in the experiment?
> We evaluate LLaMA-30b on the 2-entity CAPITALS task, under the position intervention described in section 3.2. The plotted mean log probs are averaged across N=100 samples from the 2-entity CAPITALS task.
>
> > Section 4.2: Does the experimental result really mean that the vectors that are not linear combinations of binding ID vectors do not work?
>
> No. Our experiments in 4.2 only show the converse is true: vectors that are linear combinations of binding ID vectors are also binding vectors.
>
> However, from section 4.1, the fact that random vectors do not affect binding information suggests that random vectors do not work as binding IDs.
>
> > Figure 5: What are the x and y axes? Are they the coefficients of the basis vectors?
>
> The x-axis corresponds to one basis vector. The y-axis is the orthogonalized second basis vector (i.e. $v_2 - \frac{v_2 \cdot v_1}{||v_1||\,||v_2||} v_2$). The basis vectors are plotted as black arrows in each of the heatmaps.
>
> > p. 8: What exactly is a cyclic shift?
> We use a cyclic shift to refer to a cyclic permutation. When $n=3$ as in this case, there are only two cyclic permutations --- either k -> (k+1) mod 3 or k -> (k-1) mod 3.
>
> In the experiments in fig. 6 we intervene on the context activations so that $E_0$ becomes binded with $A_1$, $E_1 \leftrightarrow A_2$, $E_2 \leftrightarrow A_0$. This can be achieved by either applying mean interventions on the binding IDs on the entities or the attributes. This corresponds to the 'entity' and 'attribute' labels in Fig. 6.
>
> > p. 2: Does a “sample” mean an example? In statistics, a sample usually means a collection of examples (data points)
>
> Yes, a "sample" here refers to sampling in the traditional sense. A sample of an $n$-entity context is obtained by sampling $n$ entities uniformly at random without repetition from $\mathcal E$ (a universe of entities), and $n$ attributes uniformly at random without repetition from $\mathcal A$ (a universe of attributes).

---

> ### Author Response · Authors · 2023-11-21
>
> > It is not entirely clear whether the authors’ conclusions are supported by experimental evidence.
>
> We are happy to discuss any question you have about our experiments and their conclusions.

---

### Official Review · Reviewer_f6eM · 2023-11-02

**Soundness:** 3 good
**Presentation:** 3 good
**Contribution:** 3 good
**Rating:** 6
**Confidence:** 3

**Summary:**

This paper studied how language models handle a linkage between entity and attribute tokens. The authors argued that language models used a component of the residual stream of each token as a binding vector to mark entities and their corresponding attributes. The authors also claimed that the binding vectors are simply added to the residual stream.

To prove this empirically, the authors presented a series of causal mediation experiment results on binding tasks, such as determining a city in which a person lives or the color of a shape based on a context. The context typically consisted of two to three entity-attribute pairs. First, the authors showed that binding vectors existed by swapping activations of an entity or an attribute. The results showed that binding changed (by means of changing the mean log probability of attributes.) On the other hand, shifting did not affect the mean log prob of the correct attributes -- suggesting that binding vectors are positional independent. The author then estimated the binding differences and showed that the authors could perform addition or subtraction to manipulate the bindings. In addition, the difference vectors are generalized across different binding tasks. But, the authors suggested one task had an alternative binding mechanism.

**Strengths:**

1. This paper presented a novel concept of binding mechanism in language models.
2. The paper provided experiment results based on many datasets, albeit toy data. Figure 3, Tables 1 and 2 supported the main claims of the binding vectors and their additivity property.
3. The paper was well-written. I found that the definitions and hypotheses were well articulated and precise. It also provided sufficient background to understand the paper.

**Weaknesses:**

**Significance of the Results**

1. While the ideas presented in this work were novel, it was unclear how generalized they are. The authors presented a series of experiments based on somewhat synthetic datasets. Had the task been reading comprehension, we might not have observed the same mechanism. I think adding more tasks did not provide meaningful results unless they required different reasoning complexities. In addition, the experiments presented in Section 3 only provided anecdotal evidence from a few samples (Factorization and Position Independence claims).
2. Although the paper presented the results using both Pythia and LLaMa as subject LMs, the results in Figure 6 showed opposite trends between the two models. I saw that the accuracies were further apart in Pythia as the sizes went up but opposite in the LLaMa cases. I think the authors should explain this. Did the binding mechanism localize to a family of models?
3. It was unclear whether there was the proposed binding ID when reasoning involving spans of tokens.

**Presentation**

1. Figure 3 shows the same image for (a) and (b). I was not sure why, or was it a mistake?
2. I found the notion of the *binding function* rather confusing in Eq (1). I had many questions about it. If the binding function specifies how an entity binds to an attribute, why is it a sum of entity representation and the binding vector? What is the relationship between the binding function and the residual stream?

**Questions:**

1. How many samples did you use to generate Figure 3? (100 or 500)
1. Were the samples you used to estimate the *differences* the same samples for testing in Table 1? If so, why?

---

> ### Author Response · Authors · 2023-11-20
>
> Thank you for the comments! We are happy that you found that we communicated the “novel concept of binding mechanism” in a way that is  “well articulated and precise”.
>
> __Presentation__
>
> > 1. Figure 3 shows the same image for (a) and (b). I was not sure why, or was it a mistake?
>
> Yes, thanks for catching this. The paper has been updated with the correct plot.
>
> > 2. I found the notion of the binding function rather confusing in Eq (1). I had many questions about it. If the binding function specifies how an entity binds to an attribute, why is it a sum of entity representation and the binding vector? What is the relationship between the binding function and the residual stream?
>
> Thanks for the feedback! We’ve updated our paper to improve clarity in section 4.1 where Eq(1) is discussed.
>
> - The main feature of the binding ID mechanism that we proposed is that instead of _directly binding_ entity to attribute, the language model _indirectly represents binding_ by binding the entity and attribute to the same binding ID (Fig. 1). The entity/attribute binding function is thus a function of entity/attribute and binding ID.
> - Binding functions are defined (Section 3) so that we can substitute the entire residual stream of either entity or attribute with the output of the corresponding binding function without changing the behavior of the LM. They should capture all information that is relevant for the LM to solve the binding task.
>
> __Questions__
>
> > 1. How many samples did you use to generate Figure 3? (100 or 500)
>
> We use 100 samples by default. We use 500 samples when estimating the means in the mean interventions used in sections 4 and 5.
>
> > 2. Were the samples you used to estimate the differences the same samples for testing in Table 1? If so, why?
>
> No, we split the dataset into a “train/test” set so that the means are estimated from the train subset and the evaluation of the intervention is done on a test set. More specifically, we split the set of entities $\mathcal E$ and the set of attributes $\mathcal A$ each into two sets, so that the contexts from the train subset and the test subset will not have any entities or attributes in common.
>
> __Weaknesses__
>
> > 2. Although the paper presented the results using both Pythia and LLaMa as subject LMs, the results in Figure 6 showed opposite trends between the two models. I saw that the accuracies were further apart in Pythia as the sizes went up but opposite in the LLaMa cases. I think the authors should explain this. Did the binding mechanism localize to a family of models?
>
> The gap is indeed evidence that Pythia is doing something in addition to the binding IDs, but note that if the binding stuff didn't work at all the baseline would be at most 33% (random), and in fact potentially should decrease as the control settings gets more accuracy (e.g. if you have 90% accuracy, an intervention that has no effect on the network should get you only to 10% permuted accuracy).
>
> Further, from the [grokking](https://arxiv.org/pdf/2309.02390.pdf) and [shortcuts](https://arxiv.org/abs/2208.11857) literature we know that LMs often learn robust and non-robust mechanisms in parallel, and may switch from non-robust to robust mechanisms with more training or at larger scales. We conjecture that the binding ID mechanism is a structured and robust mechanism that competes with an unknown non-robust mechanism. For both pythia and llama models in the range 2b-12b, the models rely significantly on the non-robust mechanism. However, as we move towards the larger llama models (13b and up), the models appear to mostly switch to binding IDs. Because 12b is the largest pythia model, it is unclear if pythia will follow llama's trend as we scale it up further.
>
> > 3. It was unclear whether there was the proposed binding ID when reasoning involving spans of tokens.
>
> Yes we agree that this is not fully answered in our work. We investigate the setting where entities are constrained to one token wide, but attributes are arbitrarily long. This corresponds to the BIOSBIAS column in Fig. 6. We found that entity mean interventions work, but to a less effective extent, whereas attribute mean interventions on the first token of the attributes do not work.
>
> __Generalizability__
>
> We are currently running additional experiments to help answer your questions about generalizability. We'll post an update soon.

---

> ### Author Response · Authors · 2023-11-21
>
> __Generalizability__
>
> > I think adding more tasks did not provide meaningful results unless they required different reasoning complexities.
>
> We agree that reasoning is an important test that we have understood the binding representations. Note that the CAPITALS task already requires a small reasoning step: the LM has to infer the capital city given the country. In our original submission we showed that our experiments preserve this reasoning ability.
>
> To further push on reasoning, we now construct a "one-hop" binding task. Here is an example from it:
> ```
> Answer the question based on the context below. Keep the answer short.
>
> Context: Elizabeth lives in the capital city of France. Ryan lives in the capital city of Scotland.
>
> Question: The person living in Vienna likes rose. The person living in Edinburgh likes rust. The person living in Tokyo likes orange. The person living in Paris likes tan. What color does Elizabeth like?
>
> Answer: Elizabeth likes
> ```
> Here, based on the binding information in the context the LM has to perform two inference steps. The first is to infer that the capital city of France is Paris. The second is to, based on the additional information in the question, infer that the person living in Paris likes tan, and output tan as the correct answer.
>
> This is a more challenging task than our other tasks, and we thus conduct experiments on LLaMA-65b instead of LLaMA-30b. Overall, we found that all of our results still hold: the representations factorize and are position independent in the same way as in the CAPITALS task, and our mean interventions also successfully switched the model beliefs as predicted by the binding ID mechanism. The results are in a new appendix (Appendix H).
>
>
> > In addition, the experiments presented in Section 3 only provided anecdotal evidence from a few samples (Factorization and Position Independence claims).
>
> To clarify, the factorization and position independence figures in section 3 show the aggregated results for 100 samples from the capitals task (i.e. 100 prompts with different names and countries filled in), and not "a few" samples.
>
> To further address "anecdotal"-ness, we have conducted the same experiments for the other binding tasks, namely PARALLEL, SHAPES, FRUITS, and BIOS. The plots are in a new appendix (Appendix J). The experiments do not show significant difference from the CAPITALS results, except for BIOS - for BIOS, attribute factorizability does not hold. This is because BIOS attributes are many tokens wide, whereas the factorizability test only substitutes the first attribute token.

---

> > ### Comment · Reviewer_f6eM · 2023-11-23
> > **Thank you for your response**
> >
> > I mentioned the complexities because it would be interesting to see how the binding ID hypothesis affects different levels of reasoning complexity (i.e., hops of reasoning).
> >
> > The authors clarified and provided additional experiments to address my concerns. I also found the "three" binding results interesting. The "binding function" confused me initially, but I found this paper well-written overall. Since the authors suggested that there were other mechanisms to "bind" the entities (in the paper and the comment about Pythia), perhaps the tone of the paper should be "We discovered one of many mechanisms." Regardless, I believe this paper has a potential and keep my recommendation.

---

### Meta-Review · Area_Chair_sfCu · 2023-12-05

**Metareview:**

The paper analyzes how LMs (specifically, Pythia and LLaMA) associates entities with their attributes described in context. They introduce a synthetic binding task, and perform causal mediation analysis, where internal activations are substituted. The paper introduces two properties, factorizability and position independence. The later chapter delves deeper, studying the nature of binding vectors (whether they are additive, or whether they have geometric relationship). The reviewers find the analysis interesting and well-motivated and interesting.

**Justification For Why Not Higher Score:**

The paper is not very polished (i.e., reviewers found multiple mistakes), and many reviewers (and AC) found the paper relatively challenging to read. I do think the paper benefit from careful rewriting based on the reviewer’s comments.

**Justification For Why Not Lower Score:**

Understanding behaviors of LMs is an important topic, and the paper proposes a well-scoped but important sub problem of entity binding.

---

### Decision · Program_Chairs · 2024-01-16

Accept (poster)